# Cryo-EM structure of the folded-back state of human β-cardiac myosin

Alessandro Grinzato[1,7], Daniel Auguin [2,3,7], Carlos Kikuti [2], Neha Nandwani [4], Dihia Moussaoui[5], Divya Pathak[4], Eaazhisai Kandiah [1], Kathleen M. Ruppel [4,6] ✉, James A. Spudich [4], Anne Houdusse [2] ✉ & Julien Robert-Paganin [2] ✉

To save energy and precisely regulate cardiac contractility, cardiac muscle myosin heads are sequestered in an 'off' state that can be converted to an 'on' state when exertion is increased. The 'off' state is equated with a folded-back structure known as the interacting-heads motif (IHM), which is a regulatory feature of all class-2 muscle and non-muscle myosins. We report here the human β-cardiac myosin IHM structure determined by cryo-electron microscopy to 3.6 Å resolution, providing details of all the interfaces stabilizing the 'off' state. The structure shows that these interfaces are hot spots of hypertrophic cardiomyopathy mutations that are thought to cause hypercontractility by destabilizing the 'off' state. Importantly, the cardiac and smooth muscle myosin IHM structures dramatically differ, providing structural evidence for the divergent physiological regulation of these muscle types. The cardiac IHM structure will facilitate development of clinically useful new molecules that modulate IHM stability.

Muscle contraction depends on myosins from class 2. In the sarcomeres, they assemble into so-called thick filaments which interact with thin filaments composed of actin to produce force during contraction[1]. Muscle contraction is energy consuming and fueled by ATP, and skeletal muscles represent between 30 and 40% of the body mass for an adult human[2]. In order to avoid energy loss between contractions, the myosin heads adopt a super-relaxed state (SRX) characterized by a low ATP turnover[3–5], promoting energy saving[3,5]. This functional state is present in all muscle fibers including skeletal[3], cardiac[5,6] and smooth[7] muscles.

While the structural basis underlying force production by myosins has been intensively studied (reviewed by[1]), the structural basis of the SRX is still being explored. From early studies on smooth muscle myosins ($_{Sm}$Myo2), the SRX has been linked to the asymmetrical

Interacting-Heads Motif (IHM) where the two heads (subfragment-1 or S1) of a myosin molecule fold back onto their own coiled-coil tail (subfragment-2 or S2)[8–11]. The IHM was further recognized from structural studies of human cardiac Myo2 ($_{Car}$Myo2[12–14],), non-muscle myo2[15,16] and muscles from animals such as sea sponges and jellyfish[16], flatworms, insects, arachnids, mollusks, fish and human[17]. From the low-resolution examined (>2 nm), all types of muscle myo2 IHM look alike and the interactions within this motif have been assumed to be conserved[16].

The role of myosin filament-based regulation to control the time course and strength of cardiac contraction has been demonstrated[18]. Hypertrophic cardiomyopathy (HCM)-causing point mutations in human β-cardiac myosin, the motor that drives heart contraction, result in hypercontractility of the heart. The 'Mesa' hypothesis[19]

[1]CM01 beamline. European Synchrotron Radiation Facility (ESRF), Grenoble, France. [2]Structural Motility, Institut Curie, Paris Université Sciences et Lettres, Sorbonne Université, CNRS UMR144, F-75005 Paris, France. [3]Laboratoire de Biologie des Ligneux et des Grandes Cultures, Université d'Orléans, UPRES EA 1207, INRA-USC1328, F-45067 Orléans, France. [4]Department of Biochemistry, Stanford University School of Medicine, Stanford, CA 94305, USA. [5]BM29 BIOSAXS beamline, European Synchrotron Radiation Facility (ESRF), Grenoble, France. [6]Department of Pediatrics, Stanford University School of Medicine, Stanford, CA 94305, USA. [7]These authors contributed equally: Alessandro Grinzato, Daniel Auguin. ✉e-mail: kmer@stanford.edu; anne.houdusse@curie.fr; julien.robert-paganin@curie.fr

proposes a unifying hypothesis for the molecular basis of this hyper-contractility wherein human β-cardiac myosin HCM mutations result in liberating heads from an off-state, putting more heads in play for systolic contraction. Thus, many HCM mutations are found on a relatively flat surface (the 'Mesa') of the myosin head, and these mutations are hypothesized to reduce the affinity of S1's putative binding partners (e.g. myosin binding protein-C or MyBP-C, titin, and myosin S2 in the case of the IHM state)[20], releasing those heads to now be involved in the contractile process[21]. The IHM-SRX state is the likely candidate for such regulation, and destabilization of the IHM interfaces has been proposed to lead to more active heads and thus increases in force[20,22–24].

Biochemical evidence has confirmed that many human β-cardiac myosin HCM mutations increase the number of heads available to engage in force production (e.g.[25,26], for review, see[20]). There is likely a class of myosin HCM mutations that affect the stability of diverse conformational states of the motor leading to different ways of modulating the force produced, rather than destabilizing the IHM state[27]. All structural studies on HCM mutations to date have relied on low-resolution (>2 nm) homology models of the human β-cardiac myosin IHM state or the β-cardiac IHM structure determined from cardiac thick filaments at 28 Å resolution[13], and a high-resolution structure of purified human β-cardiac myosin in its IHM state has been lacking.

Three higher resolution cryo-EM structures of purified smooth muscle myosin in its IHM state (SmIHM) have been reported at resolutions of 6[28], 4.3[29] and 3.4 Å[30]. These structures reveal the intra-molecular interactions within the dimer (IHM interfaces), in particular the way that the coiled-coil tail folds on the smooth muscle dimeric myosin molecule to stabilize the smooth muscle myosin sequestered state. Based on the hypothesis that the IHM is conserved across myosin types, a recent study suggested the use of homology modeling based on SmIHM to predict the effects of HCM mutations on the interfaces stabilizing the off-state of human β-cardiac myosin[28]. This study challenged the Mesa hypothesis[19] in that these SmIHM structures are significantly different from the human β-cardiac homology models, especially in the position of the S2 coiled-coil[29]. These results emphasize the urgent need for a high-resolution structure of the purified human β-cardiac myosin IHM to understand the development of inherited cardiomyopathies. Moreover, the cardiac activator Omecamtiv mecarbil[31] (OM, in phase 3 clinical trials) and the cardiac inhibitor Mavacamten[32] (Mava, approved by the FDA for treatment of HCM) have been reported to destabilize[33] and stabilize[34] respectively a sequestered state, and understanding the high-resolution structural bases for these drug effects will be important to develop new hypotheses about how to treat cardiac diseases via the development of new modulators acting on the stability of the sequestered state.

Here, we present the high-resolution structure of the human β-cardiac myosin IHM (CarIHM) solved at 3.6 Å resolution, with a highest resolution in the heads of 3.2 Å. Importantly, the comparison with SmIHM reveals major differences in the interfaces and the hinges in the lever arm, in addition to the coiled-coil region. In this work, we provide an accurate framework for studying the effects of HCM mutations, and for understanding small molecule modulator effects on the stability of the sequestered state. Our study demonstrates how the IHM structure, which is a conserved feature of all class 2 myosins, diverged in different muscle types. These structural differences substantiate the critical role the IHM plays in providing different types of thick filament regulation in these physiologically distinct muscles.

## Results

### Structure of the cardiac myosin IHM

We solved the near-atomic resolution IHM structure of human β-cardiac myosin-2 (CarIHM) via single-particle cryo-EM using a recombinant uncrosslinked protein preparation with ATP bound (see Methods and Supplementary Fig. 1). In SmMyo2 but not CarMyo2, distal

coiled-coil regions (residues 1410-1625) add to the stability of the sequestered, auto-inhibited state in addition to the S2 coiled-coil[13] (Supplementary Fig. 2). The dimeric cardiac myosin construct we used comprised two heads (human β-cardiac motor domain followed by the lever arm with human β-cardiac essential (ELC) and regulatory (RLC) light chains bound to the target HC sequences IQ1 and IQ2 respectively) and 15 heptads of the proximal S2 coiled-coil. The IHM classes used for the final 3D reconstruction represented only a portion of the picked particles (~200,000 particles over 0.5 million). We thus obtained the high-resolution structure of the sequestered off-state (IHM) of human β-cardiac myosin at a global resolution of 3.6 Å (map 1) (Fig. 1a; Supplementary Table 1; Supplementary Fig. 1). The IHM motif was immediately recognizable in the cryo-EM map, with the so-called blocked and free heads (BH and FH, respectively), the lever arm (consisting of the ELC and RLC attached to the single α-helix emanating from the globular motor domain) and the proximal S2 coiled-coil (S2). In map 1, the head regions correspond to the highest resolution and the lever arm/coiled coil to the lowest resolution (Fig. 1a; Supplementary Fig. 1; Supplementary Movie 1). By performing a focused refinement on the head/head region, we improved the map for

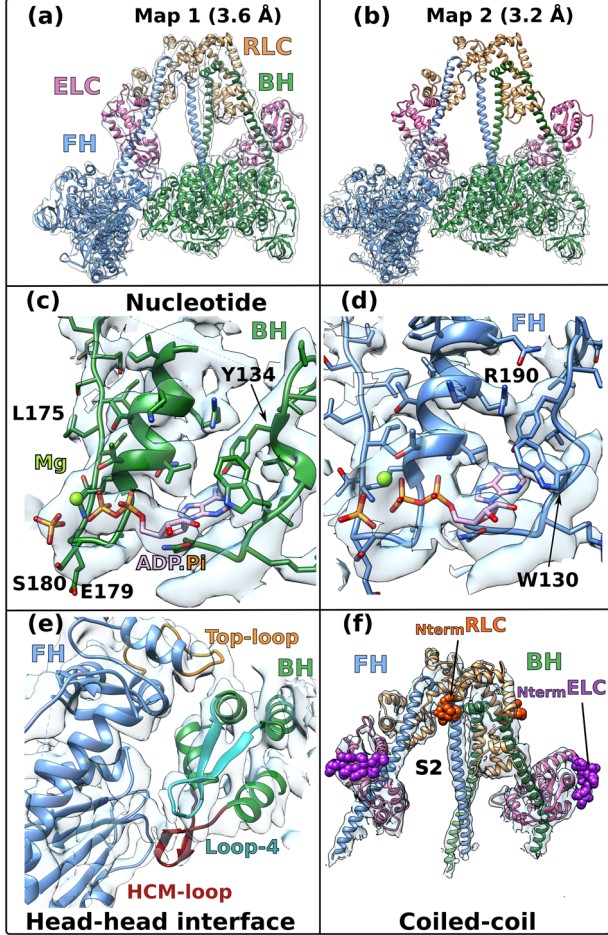

**Fig. 1 | Cryo-EM density map of the IHM of β-cardiac myosin. a** Complete map of the interacting heads motif (IHM) at a resolution of 3.6 Å (map 1). The blocked head (BH) is colored in green, the free head (FH) is colored blue, the essential light chain (ELC) is colored in pink and the regulatory light chain (RLC) is colored in light orange. **b** Map masked on the two heads improved the resolution to 3.2 Å (map 2). **c**, **d** map 2 allows visualization of the side chains of the active site, the ADP, the hydrolyzed phosphate and the magnesium. The map is represented as mesh for clarity. **e** Interface between the two heads of the IHM (map 1). **f** Light chains and N-terminal extensions (Nterm represented as spheres, NtermELC is colored in deep purple and NtermRLC is colored in orange) and coiled-coils are seen without ambiguity in map 1. Black arrows point to the N-terminal extensions.

the motor domains to a highest resolution of 3.2 Å (map 2, Fig. 1b, Supplementary Movie 2), with clear indications for the side chains (Fig. 1c). The nucleotide bound in each head is clearly ADP and $P_i$, which are trapped after hydrolysis of ATP (Figs. 1c, d; Supplementary Fig. 3a–d). Interestingly, despite the high-resolution of the map for both motor domains, some side chains are not visualized although they are likely involved in intra-molecular interactions stabilizing auto-inhibition (Fig. 1e). Flexibility clearly exists in the formation of these interactions, as shown in the diagrams illustrating the flexibility of the map (Supplementary Fig. 1f). The lability of these interactions is an important feature of the IHM structure that is required to ensure ready activation of heads when they are needed for contraction. We thus computed realistic interfaces with side chain interactions using molecular dynamics simulations (see Methods). Finally, two essential regions of regulation, the N-terminal extensions (N-term extensions) of the ELC and RLC, were partially rebuilt in density (Fig. 1f, see Methods). In contrast to the $_{Sm}$IHM structures, the density of the FH is equivalent to that of the BH in our maps, indicating that both heads are equally stabilized in the $_{Car}$IHM (Supplementary Fig. 1).

In the cardiac IHM, both heads have the lever arm primed and their motor domains are highly superimposable (rmsd 0.7 Å) (Supplementary Fig. 4a). Their structural state corresponds to a classical pre-powerstroke (PPS) state (rmsd ~0.84 Å with 5N6A[35],), defined by the structure of the motor domain from the N-terminus through the Converter ending at the pliant region (Supplementary Fig. 4b, c). The major and only difference is the larger kink in the pliant region[36] of the BH lever arm. This larger kink between the Converter and the lever arm alters the ELC/Converter interface (Supplementary Fig. 4d–f) and the so-called musical chairs (Supplementary Fig. 4d, e), a network of labile electrostatic interactions involved in the control of the lever arm dynamics at this interface[27]. Our results demonstrate that there is no significant conformational change of the Converter orientation upon formation of the sequestered state from the PPS state. The closed nucleotide pocket and the myosin inner cleft do not show any changes either (Supplementary Fig. 4g), consistent with previous spectroscopy experiments[37]. Despite its overall asymmetry, the IHM has two motor domains essentially in the same conformation.

## Previous homology models failed to predict the β-cardiac myosin IHM structure

In the absence of a high-resolution structure of the cardiac IHM, three homology models of the human $_{Car}$IHM (PDB code 5TBY;[38] MS03;[26] MA1[27]) were generated using distinct approaches, including molecular dynamics (MA1[27] and MS03[26]), and docking in the low-resolution maps of relaxed thick filaments of tarantula striated muscle (5TBY[38]; MS03[26]) or human β-cardiac myosin (MA1)[27]. However, maps obtained from negative staining of relaxed filaments isolated from striated muscle can be biased due to radial collapse of the heads and misevaluation of the radius of the fibers[39]. This bias adds further uncertainty as to the accuracy of published previous fits to negative staining data derived from striated muscle.

We compared the high-resolution structure of the IHM with the 5TBY (Fig. 2a), MA1 and MS03 (Supplementary Fig. 5a) models. Superimposition of the models with the IHM structure using the BH motor domains shows that the relative orientation of the heads strongly differs (Fig. 2b; Supplementary Fig. 5b). This difference is slightly less important in MA1 compared to 5TBY and MS03 (Supplementary Fig. 5b, c). All three models could not accurately predict the IHM interfaces in the absence of high-resolution structural information (Figs. 2b, c; Supplementary Fig. 5c). Major differences between the models and the structure also reside in the lever arm, which possesses asymmetric hinges of flexibility absolutely required to give rise to the overall asymmetric IHM structure (Fig. 2d; Supplementary Fig. 5d–f). Higher resolution is essential to correctly describe the hinges of flexibility.

Finally, the Converter orientations were compared in the three models and in the high-resolution structure. The MA1 model correctly positioned the BH and FH Converters by keeping their position as found in the single head PPS state (5N6A[35]) (Supplementary Fig. 5g). The other models positioned the Converter in a more primed position for both heads by proposing that the Relay orientation would change to further prime the Converter by 26° (Fig. 2e; Supplementary Fig. 5g). The high-resolution IHM structure now reveals that the Converter position is not modified compared to what is found in an active head but the orientation of the light chain binding region is further primed only in the BH head and comes from the hinge in the pliant region (Fig. 2d; Supplementary Fig. 5e). The MS03 model had detected the requirement of kinks in the heads and thus had proposed that the heads would adopt a so-called "pre-prestroke" conformation, with a Converter more primed than the conventional prestroke[20]. This"pre-prestroke" concept implied that slower release of products could result from these conformational changes. The high-resolution IHM structure now reveals that the slow product release (SRX) is not a result of a change in the motor conformation. In the IHM structure, both heads are in the same conventional prestroke conformation, and it is the interactions between them that slow their activity.

## The interactions stabilizing the IHM

The structure reveals five main stabilizing regions within the asymmetrical IHM: the head-head interface, the coiled-coil/BH, and three interfaces involving the lever arms: the RLC/RLC, the BH ELC/RLC and the FH ELC/RLC interfaces (Fig. 3). Supplementary Fig. 6 defines the structural elements involved in these interfaces, and the residues and nature of the interactions at these interfaces are listed in Supplementary Table 2. Both the head-head and the coiled-coil/BH interfaces involve part of a large and flat surface of myosin heads called the Mesa (Fig. 3a; Supplementary Fig. 7), which was originally defined as a large and relatively flat surface of the myosin head containing highly conserved residues among β-cardiac myosins across species, which are hot-spots for cardiomyopathy mutations[21]. Our high-resolution human β-cardiac myosin IHM ($_{Car}$IHM) structure now provides a precise definition of which Mesa residues of the BH head interact with the S2 coiled-coil, and the structure indicates that a distinct set of FH Mesa residues are also involved in a unique stabilization surface with the BH head. A subset of Mesa residues is involved in both interfaces. If both heads contain a mutated Mesa residue, as may be the case for HCM patients with the pathogenic R453C mutation, two stabilization surfaces can thus be affected, greatly destabilizing the IHM (Supplementary Table 2).

The head-head interface is formed by close interactions between the Mesa of the FH head with two BH actin-binding elements, loop-4 and the HCM-loop, as well as residues on the surface of the U50 sub-domain previously designated as the primary head-head interaction site (PHHIS)[20] (Fig. 3a; Supplementary Fig. 7; Supplementary Table 2). The structural elements of the Mesa in the FH head are the Relay helix (a connector essential for allosteric communication between the active site and the lever arm)[1], the β-bulge and HO-linker, two connectors of the Transducer[1,40] previously reported to influence myosin transient kinetics and force production (Supplementary Table 2). Interestingly, we find that the HD-linker, an element of the Transducer adjacent to the HO-linker, is also part of the FH Mesa and interacts with the BH PHHIS (Fig. 3b). In addition, the FH Converter and the ELC loop 3 also interact with the BH PHHIS. This interface includes in particular the specific $_{Converter}$Top-loop, whose flexibility and conformation greatly depends on the sequence of the Converter and its interactions with the ELC[27] (Fig. 3b). The BH loop-4 establishes interactions with the FH $_{Relay}$K503 (Fig. 3b). The coiled-coil (cc)/BH head interface involves the HO-linker and the $_{L50}$HW helix, comprising some residues of the Mesa of the BH, which both interact with electrostatic surfaces of the S2 coiled-coil while some hydrophobic residues consolidate the interactions (Fig. 3c).

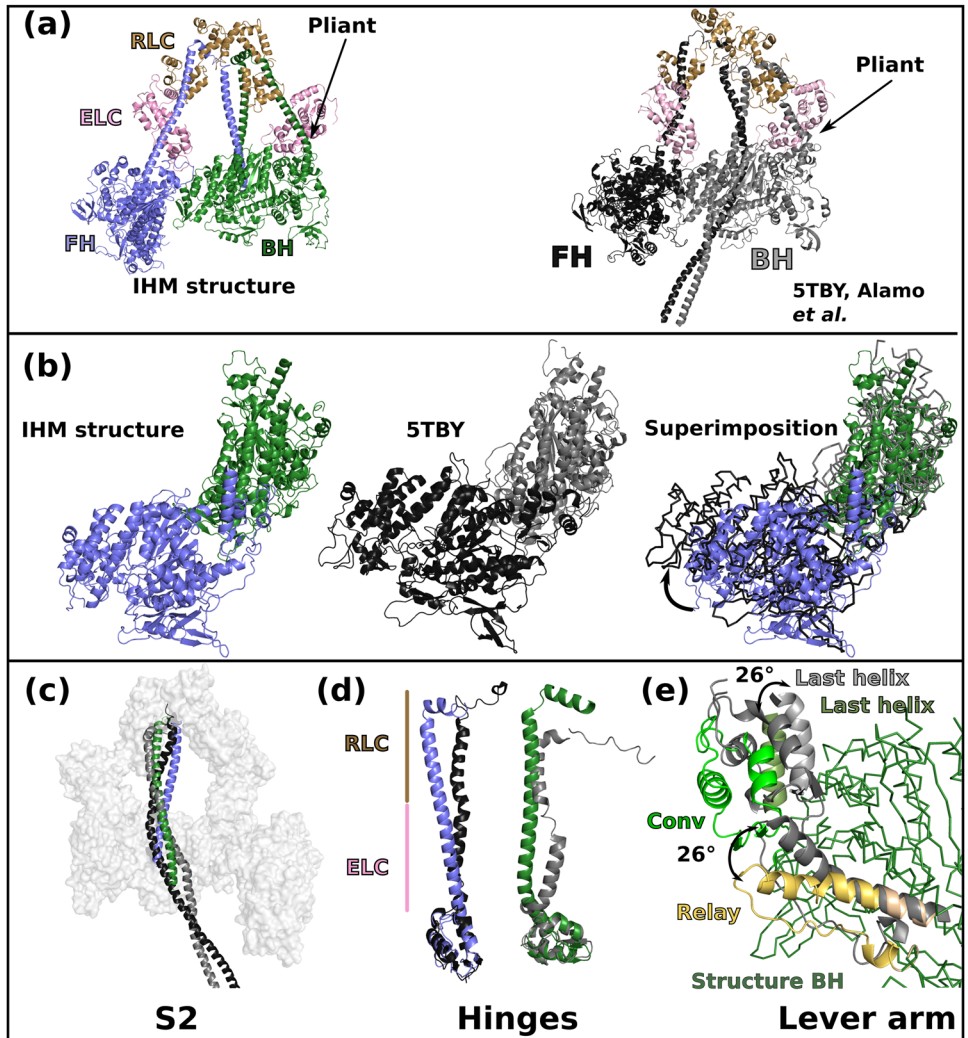

**Fig. 2 | The IHM structure strongly differs from homology models. a** Overall view of the cryo-EM structure of the β-cardiac myosin IHM (left) and of the homology model built from low-resolution maps of relaxed muscle thick filaments (PDB code 5TBY[38]). BH: blocked head; FH: free head; ELC: essential light chain; RLC: regulatory light chain. **b** The relative orientation of the two heads varies between the structure and the model. The two IHMs are superimposed on the BH motor domain (green), the difference in the FH head position (blue for the structure (left and right), black lines for the model (center and right) are highlighted with arrows. **c** S2 interacts with the BH differently in the structure as compared to the model.

**d** The lever arm conformations in the structure and the model are compared for the FH and the BH respectively, with omission of the light chains for clarity. The regions where the ELC and the RLC bind are indicated. **e** The conformation and position of the Relay and Converter greatly differ in the structure compared to the 5TBY model (grey). In (**c**), the structures are superimposed on both heads (which includes both Motor domains and the lever arms with bound LCs), in (**d**) the structures are superimposed on the Converter (residues 708-777), in (**e**), the structures are superimposed on the N-term subdomain (residues 3-202). See also Supplementary Movie 3.

The hinges and interfaces within the lever arm participate in the stabilization as well as the asymmetric orientation of the two heads of the IHM. The role of flexible hinges in the lever arm for the formation of the IHM had been anticipated[8,41,42] but they had never been described precisely. Here the model built in clear density that demonstrates that asymmetry in the various hinges within the lever arm are positioned (i) at the pliant region, located at the end of the Converter (residues 778-782) where the main helix of the lever arm is kinked by ~55° in the BH but remains straight in the FH (Supplementary Fig. 4a); (ii) at the ELC/RLC interface since differences between the two heads in the ELC loop1 conformation and in the nearby HC kink near R808 promote different electrostatic and apolar interactions at the ELC/RLC interface (Fig. 3d; Supplementary Fig. 8a, b); (iii) within the RLC lobes in which we find differences between the two heads in the positioning of the HC helices in particular for the bulky residues W816 near the inter-lobe RLC linker as well as W829 and Y823 within the N-terminal lobe of the RLC, (iv) at the RLC/RLC interface that involves direct asymmetric apolar interactions between helices 1 and 2 of the $_{BH}$RLC

and the N-term extension (M20, F21) and loop2 of the $_{FH}$RLC (Fig. 3e); (v) at the RLC/cc interface where HC residues F834-E846 adopt drastically different positions (Supplementary Fig. 8c; Supplementary Movie 3). While $_{BH}$HC residues mediate interactions within the $_{BH}$RLC up to K841 and then are part of the S2 coiled-coil, $_{FH}$HC residues interact within the $_{FH}$RLC via different contacts than those formed in the BH head, and additional stabilization contacts form after Pro838 with the surface of the $_{BH}$RLC, in particular via L839 and R845 (Supplementary Fig. 8c). FH residues form the S2 coiled-coil only after K847.

Interestingly, all these interfaces in $_{Car}$IHM are quite modest in surface area and involve several electrostatic interactions found at a distance from one another (Supplementary Table 2). In addition, most of the contacts stabilizing the IHM interface are dynamic in nature. Electrostatic interactions between long side chain residues can indeed be established with more than one partner. The concept of 'musical chairs'[27] has been proposed to describe such interfaces made of dynamic polar contacts, as described for the Converter/ELC

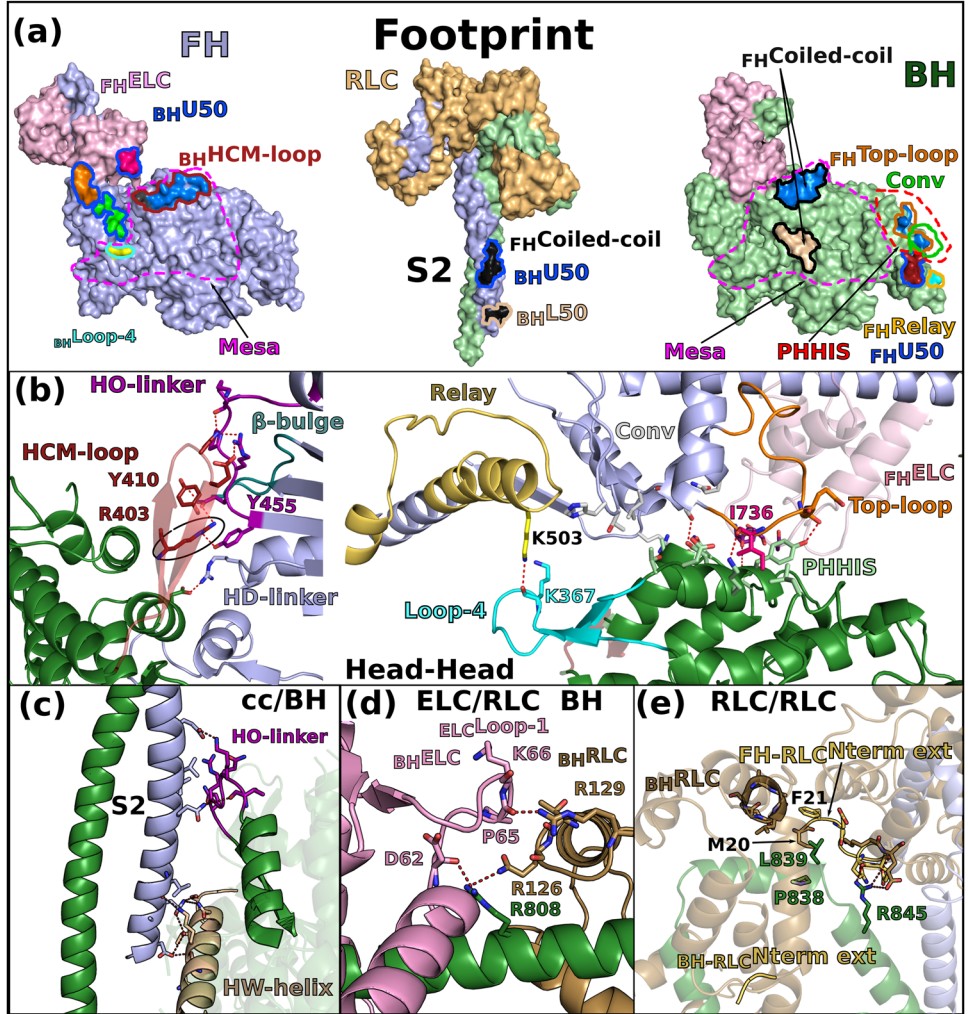

**Fig. 3 | The IHM is stabilized by multiple interfaces. a** Footprint of the interfaces stabilizing the sequestered state. The subdomains involved in the interaction are colored and the contours of interface surfaces indicate which region of the IHM is involved in recognizing this surface: L50 (wheat); U50 (marine blue); Relay (yellow); Converter (green); coiled-coil (black); HCM-loop (dark red); Loop-4 (cyan). The large flat surface called "Mesa" is delimited by magenta dotted lines. The primary head/head interaction site (PHHIS) regroups BH surface residues involved in binding the FH head next to the Mesa (see b). **b** Cartoon representation of the head-head interfaces. Key-residues involved in the interactions are represented as sticks. Red dotted lines represent the main interactions mediated by the BH HCM-loop, the PHHIS and Loop-4 residues (sticks. Key-interactions mediated by the HCM-loop through R403 are contoured in a black circle. The connectors and elements involved in the interface are shown and named in Supplementary Fig. 6. **c** Coiled-coil/BH interface. **d** ELC/RLC interface on the BH, the differences with the FH are illustrated in Supplementary Fig. 8. **e** RLC/RLC interface involving the N-terminal extension of the RLC from the FH ($_{FH-RLC}$Nterm ext).

interface[27]. Indeed, molecular dynamics has shown that electrostatic interactions at the Converter/ELC interface are labile and can arrange differently, thus maintaining a certain flexibility of the lever arm[27]. The structure demonstrates that 'musical chairs'[27] are found at different interfaces and hinges of flexibility of the $_{Car}$IHM. Such a property of these interfaces is key for allowing prompt regulation. Finally, the interfaces formed in the IHM include key-elements for actin interaction and mechano-chemical transduction (Transducer and Converter). While interactions with S2 and constraints between the lever arms efficiently result in positioning the heads away from the actin filament, blocking the dynamics of regions involved in force production also contribute to the efficient shutdown of myosin activity.

## Cardiac and smooth muscle myosin IHM differ
Importantly, the high-resolution structures of $_{Car}$IHM and $_{Sm}$IHM strongly diverge (Supplementary Fig. 9a, b). First, the $_{Sm}$BH and $_{Sm}$FH motor domains are not in a canonical pre-powerstroke conformation[30], unlike $_{car}$BH and $_{car}$FH. The conformation of the $P_i$ release tunnel[43] differs: in $_{Car}$IHM, but not in $_{Sm}$IHM, the tunnel is

closed by a conserved salt bridge formed between the switch-1 arginine (R243 in $_{Car}$IHM and R247 in $_{Sm}$IHM) and switch-2 glutamate (E466 in $_{Car}$IHM and E470 in $_{Sm}$IHM). This closed backdoor blocks the escape of the phosphate in the BH and the FH of $_{Car}$IHM (Supplementary Fig. 3c, d, 9c, d). In contrast, the positions of switch-1 and switch-2 results in an open backdoor for both heads in $_{Sm}$IHM and the critical salt bridge cannot form (Supplementary Fig. 9c, 9d). Finally, the priming of the lever arm differs in the BHs of the two IHMs. It is 10° more primed in $_{Car}$IHM (Supplementary Fig. 9e) but is more similar in the FHs (Supplementary Fig. 3e, f, 9f).

Additionally, the interfaces stabilizing the two IHMs dramatically differ. The obvious difference at low resolution for the positioning of S2 in the tarantula IHM compared to the high-resolution of the $_{Sm}$IHM had been noticed[11,29]. Yet, the fact that differences would be found in the head/head and RLC/RLC interfaces is totally unexpected since these interfaces were presumed to be conserved in all class-2 myosins in the absence of high-resolution information[11-13,16,17,28].

The unexpected head/head interface variations are related to major differences in the orientation of the two heads of the IHMs: when the

$_{Car}$IHM and $_{Sm}$IHM are aligned on the BH, the $_{Car}$FH is rotated ~20° anticlockwise compared to the $_{Sm}$FH (Fig. 4a; Supplementary Movie 4). Of note, the relative orientation of the heads in $_{Sm}$IHM is similar to the relative orientation of the heads in the homology models 5TBY and MS3, while the MA1 model is closer to the $_{Car}$IHM structure (Supplementary Fig. 10a–c; Supplementary Movie 4). At the head/head interface, the conformations of the FH β-bulge and HO-linker differ and the interactions they mediate are distinct among these two IHMs (Fig. 4b; Supplementary Table 2). In addition, the $_{FH}$Top-loop adopts different conformations in smooth and cardiac FH and distinct interactions are formed by the $_{FH}$Converter with the similar $_{BH}$PHHIS region of the U50 subdomain in the two IHMs (Fig. 4b). In fact, the sequence alignment of elements involved in the interfaces shows major differences among Myo2s (Fig. 4c), which are linked to changes in structure. For example, the Top-loop of $_{Sm}$Myo2 is one residue shorter and comprises (i) a charge reversal at the position E732 (replaced by a lysine in $_{Sm}$Myo2); (ii) replacement of the key I736 by a methionine (Fig. 4c). Such differences prevent conservation of interactions. Major sequence differences in regions involved in the head/head interfaces are also found in the HO-linker and the β-bulge. Interestingly, from sequence comparison, we can anticipate that the human IHM of skeletal muscle Myo2 ($_{Sk}$Myo2) is close to that of cardiac myosin while that of non-muscle myosin 2a (NM2a) would be similar to the $_{Sm}$Myo2 IHM. However, not all striated muscle myosins might correspond to the head/head position described here for $_{Car}$IHM. Docking in low resolution 3D EM maps of the tarantula striated myosin 2 thick filament reveals that the relative position of the heads found in $_{Sm}$IHM would be a better fit to this 3D EM map than does $_{Car}$IHM, which is also suggested by differences in sequence between tarantula and cardiac myosins (Fig. 4c).

The S2/BH head interface also markedly differs between $_{Sm}$IHM and $_{Car}$IHM, as previously noticed[29,44]. The $_{Sm}$IHM structure was solved for the so-called 10 S molecule[29,30], a form of the $_{Sm}$Myo2 molecule that does not assemble in filaments. In this $_{Sm}$IHM, interactions with two distal additional coiled-coil segments (Seg2 and Seg3, Supplementary Fig. 2), in addition to S2, strengthens the IHM. In striated thick filaments, including $_{Car}$Myo2, only S2 stabilizes the IHM. Our structure demonstrates that the $_{Car}$S2 interacts extensively with the $_{BH}$Mesa, in drastic contrast to the position $_{Sm}$S2 adopts, which only allows for a few interactions with the $_{FH}$Loop2 in the $_{Sm}$IHM[28–30]. In fact, $_{Sm}$Seg3 interacts with the $_{BH}$Mesa in $_{Sm}$IHM, in a similar way as the $_{Car}$S2 does in the $_{Car}$IHM (Fig. 4d). This raises the question of the stability of the $_{Sm}$IHM structure when $_{Sm}$Myo2 is assembled in filaments and whether the S2 would then adopt the docking site of the Seg3 in close contact with the BH head.

To precisely orient the heads and form an asymmetric IHM structure, hinges of flexibility are required, in particular in the lever arms. The structure and flexibility of these hinges can directly affect the IHM stability and it is no surprise that these hinges differ in $_{Car}$Myo2 and $_{Sm}$Myo2, considering how different the interfaces are in these two Myo2 IHMs. In both IHMs, flexibility is required at the pliant region and the kink formed is much larger in the BH head compared to the FH, which adopts a relatively straight lever arm. However, the BH pliant region is 10° more kinked in $_{Sm}$IHM compared to $_{Car}$IHM (Fig. 4e). The kinks of the HC at the IQ1/IQ2 region also differ in the two IHMs. There is an angular difference of 5° between smooth and cardiac FH heads and 30° between the BH heads respectively (Fig. 4e). The ELC/RLC interactions are thus different both for the BH and FH heads. This validates that these regions are specific ankles of flexibility required for the formation of the IHM (Fig. 4e).

Another significant difference between $_{Car}$IHM and $_{Sm}$IHM lies at the RLC/RLC interface and reveals how structural differences underlie distinct modes of regulation of the cardiac and smooth Myo2 IHM formation. Both $_{Car}$RLC and $_{Sm}$RLC are phosphorylatable in a charged and disordered N-terminal extension (N-term extension) which is not strictly conserved in sequence, although from the globular N-terminal lobe of the RLC, the position of the two phosphorylatable serines, S19 in $_{Sm}$RLC and S15 in $_{Car}$RLC, is conserved (Fig. 4f). In $_{Sm}$Myo2, the phosphorylation of RLC S19 acts as an on/off switch to disrupt the IHM and activate the muscle[45]. This was explained in the smooth IHM structures[28,30], where the $_{BH}$RLC N-term extension interacts with Seg3 and the $_{FH}$RLC N-term extension is directly part of the RLC/RLC interface. S19 phosphorylation thus disrupts both of these $_{Sm}$IHM interfaces, efficiently switching on the motor (Fig. 4f). In cardiac myosin, RLC S15 phosphorylation is not strictly necessary to activate cardiac muscle but modulates its activity[46]. The $_{Car}$IHM structure now explains this more moderate effect of RLC phosphorylation on IHM interfaces. Indeed, the RLC/RLC interface is drastically different in the cardiac IHM: the $_{BH}$RLC extension is not part of it and the $_{FH}$RLC N-term extension is found at the periphery of this interface even when the RLC is unphosphorylated (Fig. 4f). Thus, phosphorylation of the RLC would not affect the RLC/RLC interface as drastically as described for $_{Sm}$IHM and might primarily modulate the stability of the $_{Car}$IHM by interactions engaging other proteins in the thick filament. This major structural difference in how the smooth and cardiac IHM form is of interest as it provides atomic detail on how post-translational modifications may distinctly regulate myosin assembly and function in striated and smooth muscles.

Overall, the comparison between our high-resolution IHM and the only other high-resolution IHM structure solved to date provides a demonstration that major differences can be found in the structure, the stabilization, and the regulation of Myo2 IHMs which might have evolved to best serve the role of this inactive state in different muscles.

## Consequences for thick filament regulation

β-cardiac myosin, like other striated muscle myosins, is organized in thick filaments as part of sarcomeres. The tarantula striated muscle thick filament, however, differs from the β-cardiac muscle thick filament in its symmetry and organization. In smooth muscle, $_{Sm}$Myo2 is organized in side polar filaments instead of bipolar filaments[47] and the evanescence of these filaments contributes to their important adaptability. One of the major findings resulting from the high-resolution $_{Car}$IHM structure is that the interfaces stabilizing auto-inhibition strongly differ from those found in $_{Sm}$IHM. In the case of smooth muscle, the IHM is observed in a stable dimeric molecule with the distal tail stabilizing these interactions. In contrast, in striated muscles, the IHM forms on muscle thick filaments and regulation of its stability depends on interactions with other proteins found on the thick filament. Docking in low resolution EM maps of tarantula striated muscle thick filaments had suggested that the S2 position would differ compared to $_{Sm}$IHM at high-resolution[29,48]. Such docking also allows us to predict the nature of the IHM surface exposed to the thick filament and how it varies for different muscles, as illustrated in Fig. 5. This surface is found on the S2 coiled-coil side of the IHM structure. Figures 5a, b compare the relative orientation of the heads of tarantula IHM ($_{Tar}$IHM[17]) with $_{Car}$IHM and $_{Sm}$IHM when the structures are aligned on their BH. Unexpectedly, the relative orientation of the heads in $_{Tar}$IHM is closer to $_{Sm}$IHM (Fig. 5b). As a result, the difference in the relative orientation of the FH position results in a drastically different shape of the motif exposed to the thick filament in tarantula and cardiac thick filaments (Fig. 5a). Thus, different interactions with the core of the thick filament would result from the differences in the relative orientation of the heads (Fig. 5c). This unique surface of the IHM can thus differentially regulate the stability of the IHM motif via distinct interactions with the thick filament backbone and its associated proteins. Lack of conservation of this interacting surface of the IHM along evolution would result in distinct possible thick filament regulation and organization, diversifying their physiological role. Our work adds supplementary criteria for describing IHM variants that differ not only on the position of the S2[48], but also on the relative orientation of the heads. The specific features of $_{Tar}$IHM (S2 positioned as in $_{Car}$IHM and

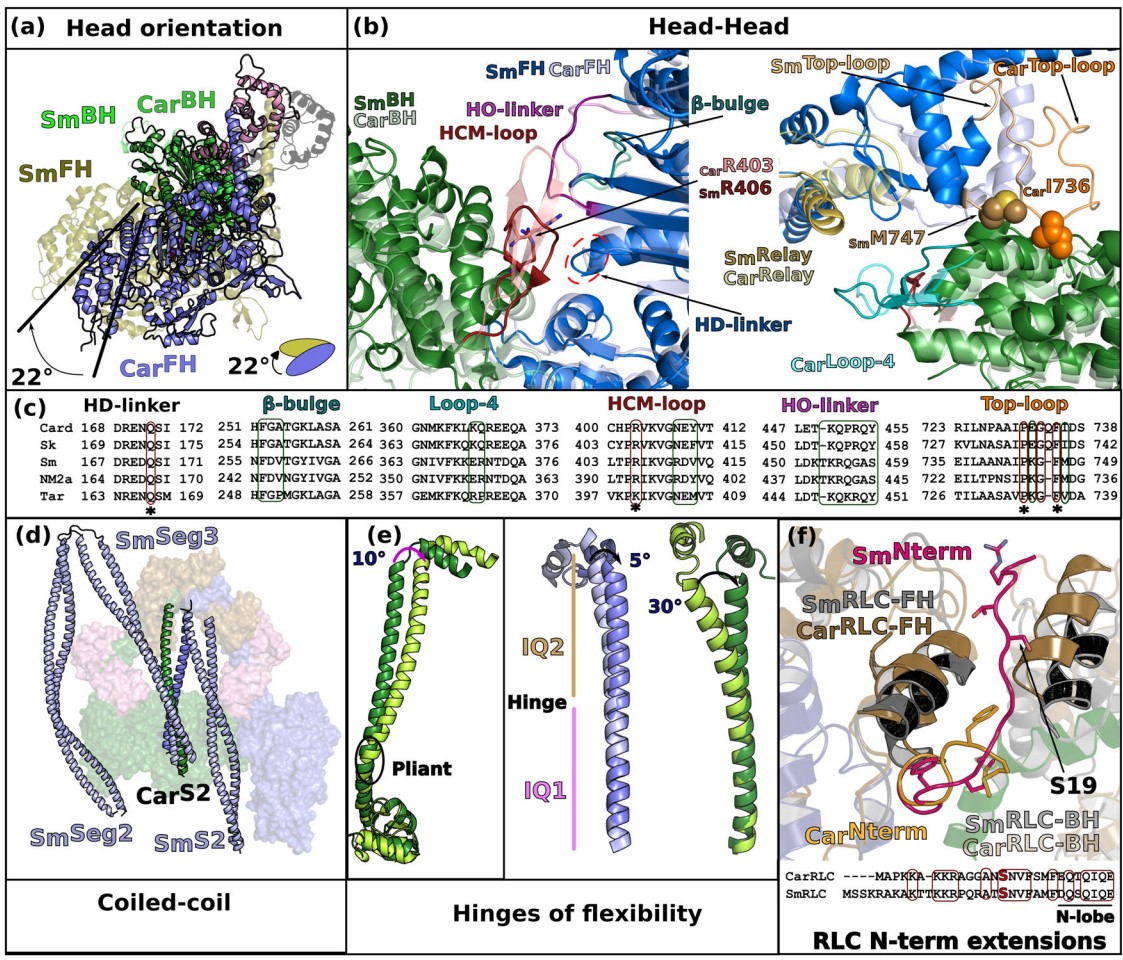

**Fig. 4 | The sequestered states of β-cardiac myosin and smooth muscle myosin (smMyo2) differ greatly. a** Relative orientation of the BH and FH heads in smIHM (PDB code 7MF3[30],) and carIHM. smIHM and carIHMs are aligned on the BH (partly shown in green), the carFH (blue) is rotated 22° anticlockwise compared to the smFH (beige), see also cartoon providing the contour of these FH heads. **b** Comparison of the head-head interfaces in smIHM and carIHM, two views are represented. On the left, IHMs are aligned on the FH motor domain; on the right, they are aligned on the BH motor domain. The carIHM is represented in transparency. Note the difference in the HCM-loop on the left, and on the FHRelay (yellow) and FHConverter positions (Blue) on the right. The red dashed line circle shows the HD-linker. **c** Sequence alignment of different structural elements involved in IHM interactions showing the divergence: Card (Homo sapiens β-cardiac myosin); Sm (Gallus gallus smooth muscle myosin); Sk (Homo sapiens fast skeletal muscle myosin); NM2a (Homo sapiens nonmuscle myosin 2a); Tar (Aphonopelma tarantula striated muscle myosin 2). Regions of interactions are in colored box: conserved in red, divergent in green. **d** Comparison of the position of the coiled-coils on the surface of smIHM heads[29] and the carIHM. **e** Comparison of the angles at the hinges of flexibility: (left) pliant region shown by an alignment on the Converter; (right) ELC/RLC interface in the FH and BH lever arms, shown by an alignment on IQ1. **f** Comparison of the RLC-RLC interface, the models are aligned on the IQ2 of the FH. A sequence alignment of the N-term extension of the RLCs is shown with conserved regions between smooth muscle RLC (smRLC) and cardiac RLC (carRLC) boxed in red. The phosphorylation sites (S19 for smRLC, S15 for carRLC) are indicated as a red "S".

orientation of the heads close to smIHM) suggest the presence of a third variant.

Interestingly, the high-resolution structure of carIHM can be fitted in the density of crowns 1 and 3 of the low-resolution negative staining density map of the relaxed filaments[13]. The resulting fit allows speculation on the elements involved in the "inter-crown" interface (Supplementary Fig. 11a). According to the model, both the N-term extensions of the ELC-FH and of the RLC-FH of crown 3 would be oriented to interact with a region containing some actin binding elements of the FH crown 1 (Loop-3, A-loop, Helix-Turn-Helix) (Supplementary Fig. 11b). According to this observation, we formulate the hypothesis that these extended regions containing several charged elements able to establish long-range interactions may concur in the stabilization of the two crowns in their respective orientation. Sequence alignment of the N-term extensions of different class-2 myosins (Supplementary Fig. 11c) suggests a specific conservation between striated muscles (skeletal and cardiac) that is not found in smooth muscle and non-muscle myosin 2 which contain an ELC with no N-term extension. The presence of the

light-chains at the "inter-crown" interface was previously suggested by low-resolution studies[12,13], but the high-resolution structure allows one to hypothesize that the presence of the phosphorylatable N-term extension of the RLC in this region may participate in the regulation, allowing control of the number of active heads during exertion by modulating the "inter-crown" interfaces. In contrast, in the helical filament of tarantula myosin 2, the crowns are oriented differently and the inter-crown interfaces differ but may also involves light chains[42,44,49] (Supplementary Fig. 11d). Future work at higher resolution is needed to confirm these concepts about how different IHMs may have evolved to fulfill the physiology of different muscle types.

## Discussion

In this work, we present the high-resolution structure of the cardiac myosin sequestered state. Until now, only homology models were available to describe the cardiac IHM[26,27,38]. This long-awaited high-resolution IHM structure allows one to visualize in detail the contacts stabilizing cardiac IHM, but also to compare it to the

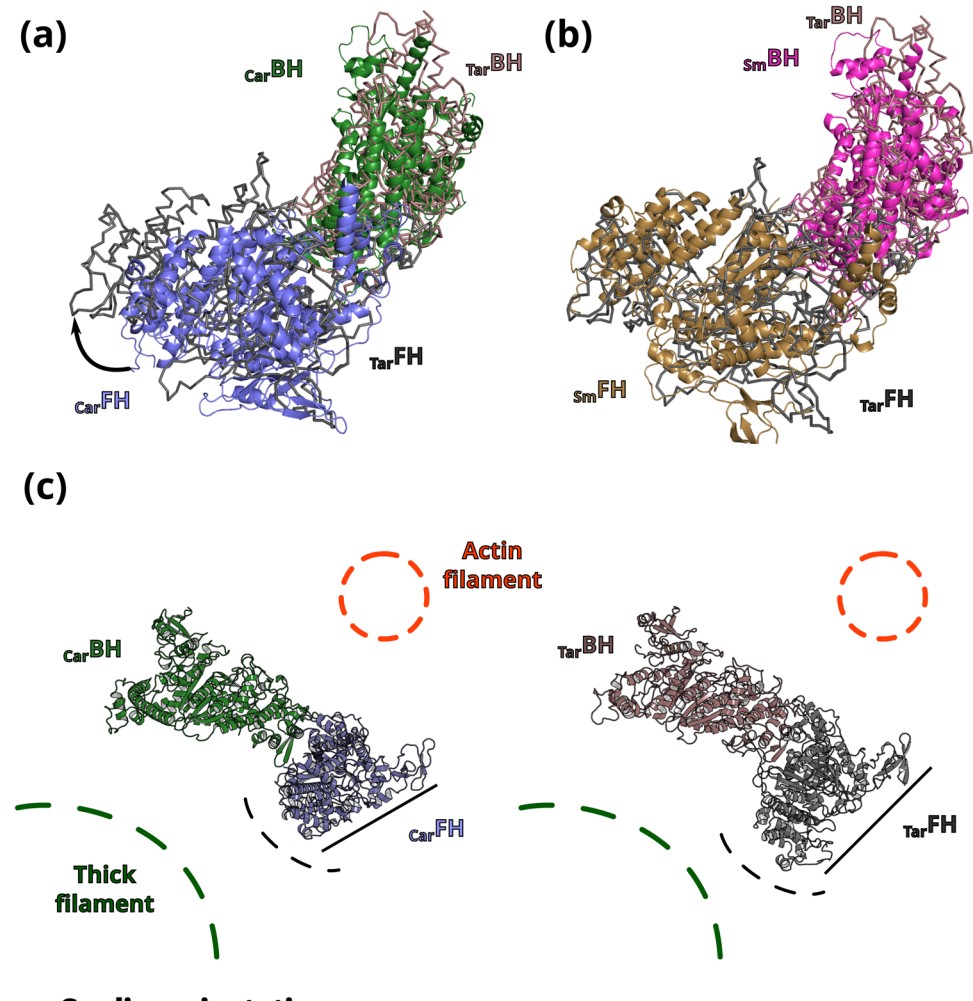

**Fig. 5 | Consequences of the differences between tarantula muscle myosin and cardiac myosin IHMs in thick filament regulation.** The relative orientation of the two heads in tarantula IHM ($_{Tar}$BH in brown and $_{Tar}$FH in dark grey) is compared to $_{Car}$IHM ($_{Car}$BH in green and $_{Car}$FH in blue) (**a**) and $_{Sm}$IHM ($_{Sm}$BH in pink and $_{Sm}$FH in gold yellow) (**b**). Interestingly, the relative orientation of the heads in $_{Tar}$IHM is closer to $_{Sm}$IHM. **c** Shows how $_{Car}$IHM and $_{Tar}$IHM are oriented relative to the core of the thick filament (dark green) and the actin (thin) filament (light red). The differences in the relative orientation of the heads would induce different interfaces. A black line shows the difference in the orientation of the FH in $_{Car}$IHM and $_{Tar}$IHM.

recent structures of $_{Sm}$IHM[28–30]. Three main conclusions arise from the present work (i) the models failed to accurately predict the $_{Car}$IHM structure; (ii) smooth and cardiac IHMs are drastically different and (iii) these structural differences highlight the molecular basis for the unique physiology and regulation of these different muscles. In addition, comparison with the tarantula striated muscle IHM exemplifies how distinct interfaces may occur amongst different striated muscles resulting in a large diversity of the stability of the IHM depending on the species and the evolution of the muscle.

The differences between $_{Car}$IHM and $_{Sm}$IHM were unexpected. While the IHM motif was shown to exist in all muscle types and organisms[16], recent work started to suggest the presence of two variants based on the position of the S2[16,48], possibly due to the ability of the distal tail to contribute to the stability of one of the variants. The $_{Car}$IHM structure demonstrates that the variability not only consists in a shift of the S2 coiled-coil but is also located (i) at the interfaces, (ii) the relative orientation of the heads, (iii) the conformation of each head, and (iv) the conformation of hinges of flexibility. Such a variability could be linked to the differences in the regulation of each fiber. The unexpected differences in the relative orientation of the heads revealed by our IHM structure indicate that the IHM can present a unique interacting surface to the core of the thick filament from one muscle to the other (Figs. 5a, b), likely facilitating distinct thick filament 'inter-molecular' interactions between crowns and with regulatory proteins such as MyBP-C and Titin. Describing these interactions awaits the high-resolution structure of the relaxed thick filament. Future structures of the cardiac thick filament can benefit from high-resolution structures of IHM solved from single particle cryo-EM to fully depict how IHM form and are regulated in situ.

The impact of HCM mutations, as well as dilated cardiomyopathy (DCM) mutations can now be evaluated with this accurate high-resolution human β-cardiac IHM structure, including those affecting the MYL2 and MYL3 genes that code for the RLC and ELC proteins, respectively. For example, the restrictive cardiomyopathy (RCM)-E143K ELC mutation[50], the HCM-R58Q[51] and HCM-D166V RLC mutations[52,53] correspond respectively to a $_{BH}$ELC residue and two $_{FH}$RLC residues of the surface of the IHM that would interact with the core of the thick filament. The IHM structure thus provides the missing puzzle piece to understand the molecular mechanism of inherited cardiomyopathies. A detailed examination of myosin HCM and DCM mutations in relation to our high-resolution structure is beyond the scope of this report and will be the subject of a subsequent publication. The availability of the cardiac IHM structure paves the road

towards the rational design of novel therapeutics against distinct families of inherited muscle disease.

## Methods

### 15-hep production and purification

The human β-cardiac myosin construct 15-hep HMM was produced using a modified AdEasy™ Vector System (Qbiogene, Inc, Carlsbad, California, USA). The 15-hep cDNA consists of residue 1 to 942 of *MYH7*, encompassing S1 and the first 15 heptad repeats of the proximal S2 coiled-coil region. This sequence is followed by a GCN4 leucine zipper to ensure dimerization, a flexible GSG linker, and finally a carboxy-terminal PDZ-binding octa-peptide (RGSIDTWV). The adenoviral vectors expressing the heavy chain myosin and the human ventricular essential light chain (ELC) (*MYH3*) containing a TEV protease cleavable N-terminal FLAG tag (DYKDDDDK) were co-expressed in differentiated mouse myoblast C2C12 cells (purchased from ATCC). C2C12 cells were infected with the adenoviral vectors 48-60 hours after differentiation and harvested 4 days post infection in a lysis buffer with the following composition: 20 mM imidazole at pH 7.5 containing 50 mM NaCl, 20 mM MgCl$_2$, 1 mM EDTA, 1 mM EGTA, 10% sucrose, 1 mM DTT, 3 mM ATP, 1 mM PMSF and Roche protease inhibitors. Harvested cells were immediately flash frozen in liquid nitrogen and stored at −80 °C.

For purification, frozen cell pellets were thawed at room temperature, and supplemented with 3 mM ATP, 1 mM DTT, 1 mM PMSF, 0.5% Tween-20 and Roche protease inhibitors before lysis with 50 strokes of a dounce homogenizer on ice. The lysate was incubated on ice for 20–30 mins to allow for the assembly of native mouse full length myosin into filaments before clarification; filament formation is promoted by the high MgCl$_2$ and low salt content of the lysis buffer. The lysate was clarified by spinning at 30,000 RPM in a Ti-60 fixed-angle ultracentrifuge rotor for 30 min at 4 °C. All subsequent steps were carried out in a cold room set at 4 °C. The supernatant was incubated on a nutating rocker with anti-FLAG resin for 1–2 h, followed by low-speed centrifugation to settle down the protein-bound resin. The lysate was decanted, and protein-bound resin was washed with >10-fold excess of a wash buffer (20 mM imidazole at pH 7.5, 5 mM MgCl$_2$, 150 mM NaCl, 10% sucrose, 1 mM EDTA, 1 mM EGTA, 1 mM DTT, 3 mM ATP, 1 mM PMSF and Roche protease inhibitors). Native mouse regulatory light chain (RLC) was then depleted by incubating the resin with RLC depletion buffer (20 mM Tris-HCl at pH 7.5, 200 mM KCl, 5 mM CDTA pH 8.0, 0.5% Triton-X-100 supplemented with 1 mM ATP, 1 mM PMSF, and Roche protease inhibitors) for 75 mins on a nutating rocker. This procedure completely removes the native mouse RLC, as indicated by SDS-PAGE analysis of the resin-bound myosin, where the band corresponding to mouse RLC is undetectable in an overloaded gel. The resin was washed with wash buffer and incubated with and N-terminally 6XHis-tagged human RLC (*MYL2*), purified from *E. coli* as previously described[26], for ~3 h. Finally, unbound human RLC was washed away, and the resin was nutated overnight in wash buffer containing TEV protease to cleave the hexameric myosin assembly from the resin, removing the N-terminal tags from both the ELC and RLC. The next morning, the protein was further purified using anion-exchange chromatography. The myosin in solution was separated from the stripped-off resin using a Micro Bio-Spin™ Chromatography Column (Bio-Rad) and loaded on a 1 mL HiTrap Q HP column (Cytiva) attached to an FPLC. The protein was eluted with a gradient of 0–600 mM NaCl (in a buffer containing 10 mM imidazole at pH 7.5, 4 mM MgCl$_2$, 10% sucrose, 1 mM DTT and 2 mM ATP) over 20 column volumes. The fractions corresponding to the protein peak were typically eluted at 200–250 mM NaCl and run on a 12.5% SDS-PAGE and pure fractions (devoid of the contaminating full length mouse myosin) were pooled. The protein was concentrated to ~10 mg/ml using Amicon Ultra centrifugal filter units, and flash frozen as 20–40 μL aliquots.

Single ATP turnover experiments from the Spudich lab (manuscript in preparation) and others[54] confirm that 15 heptads of the proximal S2 tail allow myosin to adopt the folded-back SRX state in solution. Freeze-thawed 15-hep myosin has been found to behave indistinguishably from freshly purified myosin in single ATP turnover experiments. SRX:DRX ratios are not sensitive to freeze-thawing.

### Cryo-EM data collection and processing

The sample was thawed at 4 °C and centrifugated at top speed (15,000 g) during 15 min before preparation. Note that for each batch, mass spectrometry analysis (MALDI) were performed after thawing in order to be sure that the proteins were not proteolyzed and that the N-terminus extensions were present. The sample was diluted at 0.3 mg/ml in 10 mM imidazole pH 7.5, 2 mM MgCl$_2$. 0.5 mM Mg.ATP was added and the mix was incubated 15 min at room temperature. 3 μl was applied to glow discharged UltrAuFoil 1.2/1.3 holey gold grid and vitrified in a Mark IV Vitrobot (ThermoFisher). Grids were imaged in ESRF's CM01 facility using a Titan Krios microscope (ThermoFisher) at 300 keV[55] with a K3 direct electron camera at 0.84 Å per pixel. 5154 movies were collected with 40 frames each and a dose of 0.98 e-/Å$^2$ per frame. After beam-induced motion correction with Motioncor2[56] and Contrast transfer function (CTF) estimation with Gctf[57], 4423 micrographs were selected for further particle analysis performed in cryoSPARC[58]. A total of 493,179 particles were selected after automatic picking and the first round of 2D classifications. These particles were used to generate several ab initio reconstructed 3D models that were subsequentially used for a 3D heterogeneous refinement. At the end of this first round of refinement, 213,596 particles were unambiguously attributed to the sequestered state. These particles were further processed with 3D refinement, using the cryoSPARC homogeneous refinement algorithm[58], giving final global resolutions based on the gold-standard Fourier shell correlation (FSC = 0.143) criterion[59] of 3.6 Å (map 1). To further improve the resolution of the heads region, a homogeneous refinement using a mask covering the heads region alone was carried out. This final refinement produced a 3.2 Å resolution map of this region (map 2). Local resolutions of the density maps were calculated in cryoSPARC[58].

### Model building and refinement

The head/Converter/ELC region was rebuilt based on the bovine β-cardiac myosin PPS complexed to Omecamtiv mecarbil[35] (PDB code 5N69) without solvent or ligand. The coiled-coil S2 region was fitted from the crystal structure[60] (PDB code 2FXM). The first model for the IQ2/RLC region was obtained by homology modeling with the software Swiss-Model[61] using the SmMyo2 IHM[30] (PDB code 7MF3) as a template. Each part was fitted in the density map and the model was built and optimized with the module Quick MD Simulator from CHARMM-GUI[62,63]. Steps of real space refinement were performed against map 1 with Phenix[64]. During the procedure, the head region (residues 1-781) which was at higher resolution was subjected to reciprocal space refinement against map 2 with Refmac[65]. An independent run of molecular dynamics was performed in order to improve the statistics at the interfaces and to compute realistic interactions. The system was built with the CHARMM-GUI/Quick MD simulator module[62,63]. Only the interfaces (head/head, coiled-coil/head) of the model were relaxed in a box containing explicit water (TIP3P) and salt (150 mM KCl) in the CHARMM36m[66] force field. 10 ns of simulation was performed in GROMACS (version 2018.3)[67]. A final run of real space refinement was performed against map 1. This method, combining steps of molecular dynamics minimization at the interfaces of complexes and final steps of refinement allows to compute realistic interactions for the side chains that may not be fully defined in density. The strength of this calculation routine is to avoid bias due to molecular dynamics simulations which are independent to refinement in the map and focused on the interfaces. The methodology was previously used successfully in the refinement of cryo-EM maps[68]. These steps of molecular dynamics

simulations improved the correlation coefficient between the model and the map[68]. Based on our structure, the ELC and the RLC N-terminal extensions range from residues 1–48 and 1–27, respectively, and density allowed us to attribute the conformation of the following residues: BH-ELC: 39-195; BH-RLC: 20-163; FH-ELC: 39-195; FH-RLC: 20-163. Quality indicators were calculated with Phenix, including Molprobity[69] and EMRinger[70].

## Interface analysis

The analysis of the interfaces was performed automatically with PDBsum[71]. Interactions were manually checked and extended up to 4.5 Å to discuss long range weak interactions. We chose to discuss the realistic interfaces of the IHM because we computed it with molecular dynamics. While some ambiguity exists at times in these interfaces, the electron density is good for the main chain and the large hydrophobic side-chains. As discussed in the text, most of the interfaces of cardiac IHM contain however labile side-chains (musical chairs) that can adopt more than one conformation. Lower EM density for such side chains are then apparent, even in the high-resolution regions such as the head-head regions.

## Reporting summary

Further information on research design is available in the Nature Portfolio Reporting Summary linked to this article.

## Data availability

The atomic model is available in the PDB[72] under the code 8ACT [PDB DOI]. The cryo-EM maps Map 1 and Map 2 are available in the EMDB database[73] under the accession numbers EMD-15353 and EMD-15354 respectively. The homology model 5TBY is available in the PDB. The homology model MS03 is available for download on the website of Jim Spudich's lab (http://spudlab.stanford.edu/homology-models/). The homology model MA1 is available for download as Supplementary Data 2 in[27].

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

## Acknowledgements
This work was funded by grants NIH RM1GM131981-01 (J.A.S. and A.H.), ANR-21-CE11-0022-01 (A.H.), and NIH R01GM33289 (J.A.S.). N.N. acknowledges postdoctoral funding from AHA (908934) and Stanford MCHRI (1220552-140-DHPEU). D.P. acknowledges postdoctoral funding from AHA.

## Author contributions
Conceptualization and design of the research: J.R.P., A.H., J.A.S. and K.M.R. Protein expression, purification and characterization: N.N., D.P. Negative staining: A.G., C.K., J.R.P. Set up and conditions screening to prepare the Cryo-EM grids: A.G., J.R.P. Technical support, D.M. Data collection and processing: A.G. under the supervision of E.K. Model building and refinement: J.R.P., D.A., A.H. Model analysis: J.R.P., D.A., A.H. Manuscript writing J.R.P., A.H. with the help of the other authors. Project administration: J.R.P., A.H., J.A.S. and K.M.R.

## Competing interests
The authors declare no competing interests.
