## [Peer Review File · Nature Communications]

REVIEWER COMMENTS

Reviewer #1 (Remarks to the Author):

The authors present a high-resolution (3.6 Å) structure for the beta cardiac myosin IHM. The structure is accompanied by an in-depth analysis and review for how it changes our understanding given previously lower resolution structures based on myosins from different organisms. This result is of interest to the community investigation pathomechanisms in HCM and DCM that involve MYH7 gene mutants. The description of the structure is exhaustive and outlines the differences that are observed given a higher resolution structure. The results shown offer a significant step forward in understanding the structure in this niche field. The discussion of these results may not be overly accessible to those outside of the field but this is not a major concern in this reviewer's opinion.

Comments:

The authors outline the extra insight that this structure provides. There is quite an exhaustive supplement that delves into specific mutations and rationalizes how they function. What isn't clear is what is different about the predictions of HCM/DCM pathogenicity given other previous models such as Alamo eLife. Maybe this could be clarified to guide the reader to a conclusion on how this new structure may change previous interpretations of the IHM structure. Maybe this is explained but has not been evident to the reviewer.

It would be good for the authors to also clearly state what they think the effect of having a non-full length construct of cardiac myosin may mean for the structure. Whether this provides 100% fidelity and also elaborate on the limitations of the docking into the thick filament? It would help non-experts interpret the results presented. If there are caveats it would be good for these to be openly stated. Docking goes some way to rationalizing how these results may also affect filamentous myosin.

The protein that was concentrated and flash frozen was thawed prior to beginning the cryo-EM experiments. Were the myosins tested for function prior to cryo-EM? Or were the myosins purified to remove dead heads? Or was the QC performed purely in the removal of some cryo-EM images?

Could the authors elaborate on the results in the context of RLC phosphorylation. Presumably the RLC phosphorylation level was 0% (maybe the reviewer has missed this, apologies if so). Native RLC phosphorylation is usually 0.4-0.5 mol Pi/mol RLC. This likely means that in native thick filaments there are 1 in every 2 heads with a light chain phosphorylation. How may this alter the IHM confirmation within a myosin dimer if the native state may approximate 1 head of each dimer is on average phosphorylated.

Minor:

Line 256 'of the IHM in cardiac.' Should be "cardiac myosin'?

The abstract is quite non-specific and doesn't really state what the findings are, given that this is outlined very neatly in sub-headings within the manuscript maybe this can be translated to the abstract to capture the headline findings more specifically.

Very minor:

Some of the prose is quite dense and the sentence structure is often slightly confusing. This doesn't significantly impact the understanding of the manuscript but a little tightening would go a fair way in making this manuscript easier to read/understand.

Reviewer #2 (Remarks to the Author):

In this paper, Grinzato et al. use two-headed cardiac myosin II fragments expressed in, and purified from, C2C12 cells to obtain cryo-EM structures at near-atomic resolution of the interacting head motif (IHM) of myosin. This motif has the two myosin heads folded back onto themselves forming one, so called, free head (with unblocked actin binding site but low ATP turnover rate) and one blocked head where the latter also has fairly strong interactions with subfragment 2 (S2) of the two-headed myosin. The IHM is believed to be the structural counterpart of the biochemically defined superrelaxed state that exhibits 10-fold lower basal ATPase activity than free S1 in solution. Compared to earlier model studies, the work provides different structures and appreciably more detailed pictures of the interactions between the blocked and free head as well as between the heads and the S2 domain.

The findings are of relevance for understanding both the mechanisms for the formation of the IHM and its functional consequences. However, the results are also of interest from a translational perspective because several disease causing mutations and recently developed cardio-active drugs are believed to increase or decrease the propensity of myosin to form the IHM. In deciphering the detailed interactions behind the IHM in cardiac myosin II, the authors detect several differences from the interactions seen in other myosin II isoforms (based on earlier studies). They therefore go on to claim that their findings challenge the recent suggestion (Lee et al., PNAS 2018) that the IHM has been evolutionarily conserved in myosin II since before the origin of animals.

The paper is generally well and clearly written (but can be improved in some respects as suggested below). The motivation behind the study is well laid out in the Introduction with generally adequate citation of the literature. Likewise, I judge that the methods are sufficiently well described to allow repetition of the experiments and analyses. Presumably the Methods should be moved to the main paper.

The results provide several novel insights of critical importance for understanding the IHM and its functional importance in health and disease. One example in physiology is how the different inter-domain interactions between cardiac and smooth muscle myosin relate to different roles of the IHM in regulation of smooth and cardiac muscle contraction. Another example is how the structure of the IHM allows the model for the cardiac thick filament to be optimized with enhanced understanding of functional peculiarities. With regard to pathologies and myosin pharmacology the results elucidate, in appreciable detail, (e.g. Supplementary Table 2 with related figures and text) the mechanism by which cardiomyopathy causing mutations and drugs might modify the IHM. Generally, the conclusions and claims made are well supported by the provided data.

Overall, I judge the study to be of high class methodologically, with well designed and carefully conducted, experiments and analyses with results that lead to a new level of understanding of the IHM in cardiac muscle. This understanding spans from individual intermolecular interactions to the filament structure. The paper addresses issues of great importance both fundamentally and translationally, e.g. for understanding of diseases and drug mechanisms, opening for new developments in therapeutics. Based on my generally favorable assessment, I strongly recommend the paper for publication. However, some issues, outlined below, require consideration.

Specific points

1. Clarity of presentation.

Whereas the presentation is rather clear overall, there are a number of issues that can be improved, particularly to enhance the accessibility of the study to researchers outside the field of structural biology.

i. Figure 1, panels b, c

- Please increase the space between panels b and c
- Pi could be colored in the text the same way as the phosphates in the structure.
- Some key amino acid residues could be labelled to facilitate navigation

ii. Figure 3

- A line in panel b (right) from text "Car Top loop" to the structure would remove any remaining ambiguity
- Increase font size of sub-script Sm and Car for right part of panel f

iii. Extended Data Fig. 1.

- Too small font size in panels f and g.

iv. Extended data Fig. 4

- Panel b. Would it be possible to orient the structures the same way as in panel a and also indicate the region observed in b within a box in a?

v. Text - general

- I noted an inconsistency in use of subscript for "BH" and "FH" e.g. "BHRLC" vs "BH RLC" etc. (e.g. text on lines 195 and 199). Please check for consistency throughout.
- Line 289: "relaxed muscle thick filament" -> "relaxed striated muscle thick filament". Considering that smooth muscle is also a muscle.
- Line 303: Please define "MyBP-C" upon first use.
- For increased clarity and readability, I suggest that the authors move Extended Data Figures 3-4 to the main paper as these figures are quite extensively discussed in the main text.
- Finally, this may not be a problem but I just remind the authors that most color blind people have problems distinguishing red and green.

2. Other issues

i. Lines 297-298: "increases cross-bridge kinetics" I am not sure exactly what is meant by this phrase. While I am not a native English speaker, I would prefer a more specific term such as "increases the rate of cross-bridge attachment-detachment", "increases the rate of cross-bridge turnover" or something similar. A possible less specific alternative would be "leads to faster cross-bridge kinetics".

ii. Lines 75-76. The authors should consider to further weaken the statement "...challenging the claims that the IHM are conserved amongst the different myosins 2" (see also Abstract). Whereas the detailed mechanisms for the IHM seem to be different, I am not sure that the present data support the idea that IHM, as such, is not conserved amongst different myosin 2 isoforms.

iii. The proteins used were expressed in C2C12 cells in a highly controlled and elegant way with both the myosin heavy chains and the light chains from fast ventricular beta-myosin

and all derived from the human genome. Nevertheless, the proteins are in some respect artificial by not being derived from cardiac muscle cells. This should be briefly discussed. For instance, could the use of leucine zippers to stabilize the dimerized state in any way affect the IHM structures? Would a different posttranslational modification pattern than in human cardiac cells be of relevance? I presume that this pattern is different in the C2C12 derived proteins? E.g. I presume that the regulatory light chains are not phosphorylated whereas some level of phosphorylation is generally seen in the cardiac cells. What about possible arginylation (e.g. see Rassier and Kashina, Am J Physiol 2019), methylation etc? Please clarify and discuss.

iv. Lines 111-112 and lines 140-141. These bits of text seem to contradict each other related to priming of the lever arm. Please clarify!

v. Lines 103-104. The authors mention Movies but then refer to Extended data Fig. 1f?

Reviewer #3 (Remarks to the Author):

This is an exciting paper that reports for the first time the near-atomic structure of the interacting-heads motif (IHM) for human cardiac myosin. This has been a key goal of the cardiac muscle field for several years as it reveals critical intramolecular interactions between myosin heads and the heads and the tail that create the IHM and underlie key aspects of cardiac muscle physiology (the super-relaxed state, SRX). Previous studies had approached this goal by studying IHMs from other species, but at low resolution, and by high resolution studies of a smooth muscle IHM, which turns out not to be the same as cardiac muscle. Solution of the cardiac IHM at near-atomic resolution reveals in detail the interactions involved in the motif, the locations of some mutations that give rise to HCM and DCM, and provides a basis for interpreting and designing drugs to treat these diseases. This breakthrough will have a major impact on the field. However, the paper needs substantial work before publication. This includes clarification of many points, correcting descriptive errors, citing key papers that have led to this work that are currently ignored, improving the English, removing overinterpretation and simplifying description of the results. Fixing the points below I believe will turn this beautiful piece of work into a classic paper.

Major points:

1. Some key references are missing or not used appropriately. The work of Al-Khayat 2013 on cardiac thick filament structure [ref 10] is cited at many points while that of Zoghbi PNAS 2008 – which preceded Al-Khayat by 5 years – is missing. Zoghbi made the key breakthroughs in understanding vertebrate thick filament structure, which Al-Khayat then followed, and came to the same conclusions. Please properly cite this pioneering work when citing Al-Khayat. Credit should also be given to Gonzalez-Sola, BJ 2014. Nag 2017, on the location of HCM mutations is cited, but the parallel work of Alamo 2017 (eLife), coming to similar conclusions, is omitted. Please cite appropriately. Burgess 2007 was the first to reveal the IHM together with the course of the tail, in studies of smooth muscle myosin, and should be cited (L48). L47 – would be helpful to add the review by Craig/Padrón [31] that addresses the relation between IHM and SRX. L48: the first to show details of nonmuscle myo2 IHM was Jung MBC 2008, which is not cited. L64-65, first mention of mava and OM should quote Green Science 2016 and Malik Science 2011. L76: "SmMyo2 IHM structure would have been a good model to discuss the cardiomyopathy mutations." Scarff 2020 and Heissler 2021 made these claims but Yang 2020 did not and should be excluded from this statement. L80: similar issues were discussed in Craig/Padrón [31], which should be cited.

2. L268-307. This section (Consequences on muscle physiology) and related discussion should be greatly curtailed, or omitted, as it is an over-interpretation of the data. The

authors imply as much in L66-68 (“The sequestered state of β -cardiac myosin has been described at 28 Å resolution¹⁰, but that is insufficient to describe precisely the IHM interfaces.”). But they then go on to do just that – fitting their high-resolution structure into the 28 Å-resolution cardiac map. Clearly, even with their beautiful high-resolution IHM, there is a lot of wiggle room when placing it in a low-resolution map, so interpreting interactions between crowns at atomic resolution is pretty much guess work. What is needed is a high-resolution map of the filament. This would be a point worth making. But the current attempt to interpret inter-crown interactions on the basis of a 28 Å-resolution map is spurious, dilutes the high-resolution interpretation that is possible within the IHM, and could easily be misleading to the field. Another critical point in this regard is that the Zoghbi and Al-Khayat reconstructions were both based on negative stain data. A paper published earlier this year (Koubassova BJ 2022) shows that the IHMs in these reconstructions have collapsed roughly 35 Å radially onto the filament backbone. With such a major (artifactual) change in structure, interpreting interactions at atomic resolution is clearly fraught with danger, and would lead readers to conclude that the level of detail known is much greater than it is. The authors should omit this section of the paper along with Figs 4, 5a and ED Fig 9. See minor points below for other concerns with this section.

3. An important point, not discussed, is acknowledgment that the IHM solved here may not be the same throughout the thick filament. It is well accepted that IHMs in the C-zone are better ordered and have therefore been the basis of the filament reconstructions (Zoghbi, Al-Khayat). But what of IHMs in the P- and D-zones? Do the heads even form IHMs in these regions? If so, are they the same as the IHM solved here? These subtleties would be worth pointing out in the Discussion.

4. Fig 4. There are a number of minor points that would need to be corrected in the figure and legend. But, as stated, this figure should anyway be omitted. Likewise Fig. 5a and ED Fig 9, as they imply a resolution of this interface that has not been achieved.

5. ED Fig 1. The image quality/resolution needs to be much better. The figure quickly becomes pixelated when enlarged. 2D classes are too small and no secondary structure is visible. “(e) Shows the flexibility within the different classes used to reconstitute the map.”: (e) should be (f), and “local flexibility” should be explained. The legend says two orientations, but 3 are shown. What are those orientations – explain in relation to the 3D map? “(f) plots the different orientations of the particles.”: “f” should be “g”.

6. L106. Please show representative examples of cryo-EM densities of the interaction sites of the side chains to give the reader a clear idea of how well the predicted interactions are actually seen in the density map. Looking at the fit of the PDB model to the map seems to show substantial regions where there is empty density and also density outside the map – can the fitting be improved? An example is the fitting of ADP.Pi to the map, which does not look unique.

7. L84. The construct is stated to be native, which suggests coming from tissue. It is recombinant.

8. Methods (ED p11). The construct is stated to be human, but involves replacement of mouse RLC with human RLC. Some quality control would be useful to show: how much was exchanged, how much was the mouse RLC depleted? It would help to provide a cartoon of the vector map with the construct used to express the recombinant protein. Is the FLAG tag only attached to the ELC, not the HC? Not clear. It would also help to have a figure showing the ion-exchange profile to give an idea of the salt concentration for the protein elution. For homology modeling, what method/program was used? Reference? “While some ambiguity exists at times in these interfaces, the electron density is good for the main chain and the

large hydrophobic side-chains". That means some side chains reported are based on simulated models, not side chain density? Please clarify. Why was "bovine β -cardiac myosin PPS complexed to Omecamtiv mecarbil2 (PDB code 6N69)" used for model building instead of 5n6a (bovine) or 4p7h (human cardiac) motor domain? Note that, anyway, it should be 5n69, not 6n69 (Fig 5 legend, Model building, and ED Table 1).

9. L124. The correct ref for 5tby is not [27], but Alamo eLife 2017. One could question whether it is more appropriate to use MS03 in the comparison (not validated and not in the PDB database) rather than 5tby, which is. 5tby should be added to the comparison in ED Fig. 4a as it is the only deposited human cardiac IHM PDB. A confusing point: L142-143 states that in MS03 the heads would be in a pre-pre-stroke conformation. They also say that 5tby and MS03 are similar enough to study only one (MS03) (L124-125). And yet according to Alamo 2017 eLife, 5tby was thought to be in a pre-stroke conformation (not pre-pre). This appears to be self-contradictory, and further suggests that 5tby should also be compared.

10. L149-213. I found this an extremely detailed and complex description. It would be very helpful if it could be simplified/reduced to make it more accessible to the reader. Likewise ED Table 2. As it is, this section will not be very clear or accessible to a non-specialist reader.

11. L218. The refs here should be [11] and [12], not [31]. Also Yang [14] pointed out a major difference between the filament IHM and that in SmMyo2, in that S2 is shifted by 20 Å compared with the IHM in the thick filament (and replaced by seg3). This is a major difference, making it clear that IHMs are not all alike, when seen in detail. It seems that the authors may be aware of this as they state (L62-64): "Interestingly, the recent SmMyo2 IHM structures challenges the Mesa hypothesis¹⁴ in that these structures are significantly different from the human β -cardiac homology models²⁰." Yang [14] noted this early on and should be acknowledged on L218.

Minor points:

1. Most of the paper is pretty well written, but there are lots of non-idiomatic English expressions. Please have one of the native English-speaking authors correct these to make the work as clear as possible to readers.
2. L25: interacting-heads motif (hyphenated, heads plural).
3. L49. Worms should be flatworms (to distinguish from earthworms).
4. L51. Certainly, the isolated motif of interacting heads has been conserved [refs 11, 12] as well as in muscle thick filaments [ref 27], even if the details vary as shown in the current paper, and previously [ref 14].
5. L85-86. I found this description of the section of coiled-coil (residues 1410-1625) slightly confusing as it is just a middle stretch of tail, not the entire length of the tail beyond S2. One has to look carefully at the SmMyo2 papers to see which bit of the tail they mean. Maybe indicate the region on ED Fig 2, or omit ED Fig 2 as it is not really necessary, having been shown in the previous SmMyo2 papers.
6. L90-91. "Only a portion..." Is this based on 3D classification? If so, what other structures were found?
7. L99: How well does the post-hydrolysis model correspond with the map? What is the strong density near phosphate 1 of ADP in the FH? The adenine portion does not follow the density well and there is unmodelled density near the benzene ring in the BH. Overall, the fitting shown in Fig. 1c-e (and in the fitted 3D maps) does not look so impressive. Can it be improved?
8. L103-104. Movies of 2D classes are mentioned but not shown. ED Fig. 1f and 1g needs to be properly explained.
9. L106. Should be molecular dynamics simulations

10. L107. Please define the ELC and RLC N-terminal extensions. Which residues? References for each?
11. L108-109. Both heads have similar stability. This is surprising based on previous studies. Could it be because the 200,000 molecules in the reconstruction were specifically included because they gave a good match to the fully formed IHM (with BH and FH in their correct positions)? What maps did the 300,000 unused particles generate? For example, did any of those maps show a blocked head but not a free head (disordered?), as one might have expected? If so, this could support the concept of FH mobility. See also Minor point #6 above.
12. L112-113. This (pre-powerstroke state) appears to differ from Heissler for SmMyo2, who proposed a different state, off the main crossbridge cycle. This could be an interesting discussion point, given that smooth myosin is switched off 10X more than cardiac, which Heissler explains by a different, more inhibited, structure from a classical pre-powerstroke state. Is RMSD based on the C- α atom or include the side chains?
13. L115-116. The larger kink was seen in all 3 SmMyo2 structures solved, also in past Alamo papers and in Wendt 2001. Reference to these would be appropriate. Same for L187-188.
14. L145-147. A comparison with Heissler SmMyo2 structure would be informative here.
15. L166. Ref for PHHIS?
16. L203-204. Their relative position is distant – not completely clear. Is the meaning that the patches are at a distance from each other?
17. L205-206. Most of the contacts made are dynamic in nature. Evidence, reference?
18. L208. It would help if the authors provided a short intuitive description of what is meant by “musical chairs”.
19. L231-233. This is no surprise, given the well-established similarities between smooth/nonmuscle which are different from the similar cardiac/skeletal.
20. L237-240. These facts (“Surprisingly, the cardiac S2...”) were clearly anticipated by Yang [14], which should be referenced. Their ED Fig 6 is similar to Fig 3d here. Scarff and Heissler did not discuss this.
21. L245-246. It’s not clear from Fig 3e that S19 and S15 positions are conserved. Only S19 is shown.
22. L257. “Underlies” sounds like a statement of fact. But it is an interpretation.
23. L277. Refs 35-37 are inappropriate. Huxley and Brown [35] discovered the perturbation, but did not know anything about it structurally. Luther [36] doesn’t address the issue of interaction between crowns. And Al-Khayat [37] is an early and inadequate reconstruction to answer this question. The two appropriate references to quote would be Zoghbi 2008 and Al-Khayat [10]. However, this is part of the section that we suggest should be deleted anyway (Major point #2).
24. L294-295. The inter-crown interaction in tarantula does involve the RLC. It appears not to in Fig. 4d because the authors have placed the cardiac IHM into the tarantula map. But that is inappropriate. When the tarantula IHM is placed in the tarantula map, there is interaction involving the RLCs. This is because the tarantula RLC has a much longer N-terminal extension than cardiac (Alamo 2008, Brito 2011, Alamo 2016). This incorrect reasoning (and several mistakes in the legend of Fig 4) is another reason that this figure and the section Consequences on muscle physiology should be removed.
25. L313. The authors quote refs 21, 27, 28 and 40. They omit reference to Alamo eLife, a parallel study to Nag 2017 [28]. Instead they refer to Alamo Biophys Rev 2017, a review of the filament IHM—which should be quoted in the IHM introductory section. I am guessing this is simply a mix-up of references. The eLife paper should be quoted, but is currently not in the reference list at all. The same on L315. Similarly, L322-323. Quote Nag 2017 and Alamo 2017 (eLife) who first suggested this.
26. L337-339. As stated earlier, reference to inter-crown interactions should be toned down or removed (major point 2 above). L338. Fig 5a is also overinterpreted for the same reason and should be omitted. L342-343: Again premature interpretation of inter-crown interface,

- when we really do not know its structure at near atomic resolution. Similar for L345-349.
27. L345 and Fig 5a. These DCM mutations and the figure and legend are very confusing. Just one example – E525 is implied to be in the inter-crown interface (but doesn't appear to be so at all) and as destabilizing the IHM (L582-583). The reasoning and statements have little support. In fact E525K has been shown to stabilize, not destabilize SRX (Rasizzi BioRxiv 2022), and the mechanism appears to have nothing to do with the inter-crown interface, but in fact to stabilization of the BH/S2 intramolecular interface. Again, the inter-crown section should be deleted. One might add that the mutational analysis in general should be improved, as some of the mutations discussed are not known to be pathogenic and therefore unlikely to be structurally significant (cf. Alamo eLife, which analyzed only pathogenic or likely pathogenic mutations).
 28. Fig 1f. It might help to have a different color for the NTEs to make them stand out.
 29. L518-519. The phosphate is hydrolyzed? The fitting in (c) and (d) is not convincing. It could help to make a figure showing only the map density of the ligands, ions and surrounding/interacting side chains and the corresponding parts of the fitted models and remove the rest. This would make things clearer to the reader. This is done nicely in Heissler Fig 1.
 30. L528. Purple should be magenta.
 31. L539. The relative positions of FH and BH are not clear from (a). Please change the representation.
 32. Some of the domain/loop terminology is not well known (e.g. what is top loop?). It would be very helpful to have a table with such definitions, the amino acids involved, etc. to make navigating the figures easier.
 33. L549-550. Yang [14] should be cited, as they showed this there.
 34. ED Fig 3. "(e) and (f) compare ... the musical chairs are represented on both panels." How are they represented? What is the color coding for this figure?
 35. ED Fig 7. What is the color coding?
 36. ED Fig 8. I can see how S2 matches up to the BH mesa. But the labeling on the FH and BH interaction surfaces does not match up. Please fix. Fig 2a footprint does a better job using colors. What is the color coding for the surface charge in ED Fig 8.
 37. ED Movie 1 and 2 legends. Colors used to define BH and FH (blue and green) are reversed.
 38. ED Table 1. Map sharpening B-factor (\AA^2)?

REVIEWER COMMENTS

Reviewer #1 (Remarks to the Author):

The authors present a high-resolution (3.6 Å) structure for the beta cardiac myosin IHM. The structure is accompanied by an in-depth analysis and review for how it changes our understanding given previously lower resolution structures based on myosins from different organisms. This result is of interest to the community investigation pathomechanisms in HCM and DCM that involve MYH7 gene mutants.

The description of the structure is exhaustive and outlines the differences that are observed given a higher resolution structure. The results shown offer a significant step forward in understanding the structure in this niche field. The discussion of these results may not be overly accessible to those outside of the field but this is not a major concern in this reviewer's opinion.

We thank Reviewer #1 for the positive feedback regarding the manuscript. We have taken into account the remarks to improve the accessibility of the manuscript to a more general audience while also retaining a level of detail that will satisfy specialists in the field.

Comments:

The authors outline the extra insight that this structure provides. There is quite an exhaustive supplement that delves into specific mutations and rationalizes how they function. What isn't clear is what is different about the predictions of HCM/DCM pathogenicity given other previous models such as Alamo eLife. Maybe this could be clarified to guide the reader to a conclusion on how this new structure may change previous interpretations of the IHM structure. Maybe this is explained but has not been evident to the reviewer.

The model from Alamo *et al.*, eLife, 2017 is a predictive homology model based on the fit in a low-resolution map (28 Å). There is no possibility at such resolution to be sure of the exact conformation of the motor domain or the structural regions involved in the interface between heads, and the model cannot describe side chain interactions. The model could propose which mutations among those previously described as causing HCM and DCM would likely belong to structural elements predicted to be close or engaged in interactions; however, the rough model was wrong in describing the interfaces as well as the exact structure of the heads, in particular the hinges. As an example, the previous model incorrectly predicted the interface between the ^{FH}Converter and the ^{BH}Motor domain (see Supplementary Fig. 4).

In contrast, our structure directly shows the residues that are engaged in the interfaces and those that stabilize the conformations of the elements involved. It also provides, for

the first time, an understanding of how mutations in internal regions of the motor, including in the hinges such as those found in the lever arm, could impact the stability of the IHM. To make this point clear, we have added a specific section where we compare specifically the previous homology models with the structure at high-resolution: **“Previous homology models failed to predict the IHM structure”** (l. 147-195). The model 5TBY is compared to the structure in **Figure 2** and the other available models are compared with the structure in **Extended Figure 5**. The additional paragraph now indicates that the models are not appropriate to predict the precise impact of the mutations since they all failed to predict (i) the relative orientation of the heads; (ii) the conformation of the head and the lever arm; (iii) the conformation of the S2 coiled-coil.

In order to satisfy Reviewer #3’s comment, we have removed **Extended Data 1** and the paragraph **“Inherited cardiomyopathies and therapies”** from the manuscript. Instead, a paragraph was put in the Discussion:

“The impact of HCM mutations, as well as dilated cardiomyopathy (DCM) mutations can now be evaluated with this accurate high-resolution human β -cardiac IHM structure, including those affecting the MYL2 and MYL3 genes that code for the RLC and ELC proteins, respectively. For example, the restrictive cardiomyopathy (RCM)-E143K ELC mutation⁴⁹, the HCM-R58Q³³ (Kampourakis et al., 2018) and HCM-D166V RLC mutations^{50,51} correspond respectively to a _{BH}ELC residue and two _{FH}RLC residues of the surface of the IHM that would interact with the core of the thick filament. The IHM structure thus provides the missing puzzle piece to understand the molecular mechanism of inherited cardiomyopathies. A detailed examination of myosin HCM and DCM mutations in relation to our high-resolution structure is beyond the scope of this report and will be the subject of a subsequent publication. The availability of the cardiac IHM structure paves the road towards the rational design of novel therapeutics against distinct families of inherited muscle disease.” (l. 441-451).

It would be good for the authors to also clearly state what they think the effect of having a non-full length construct of cardiac myosin may mean for the structure. Whether this provides 100% fidelity and also elaborate on the limitations of the docking into the thick filament? It would help non-experts interpret the results presented. If there are caveats it would be good for these to be openly states. Docking goes some way to rationalizing how these results may also effect filamentous myosin.

Functional data comparing the DRX/SRX ratio between the construct used (15hep) and longer ones (human β -cardiac containing 25 heptads (25hep) HMM [Nag et al] or bovine cardiac HMM [Rohde et al]) did not show any significant difference and is consistent with our conclusion that no artifact in the IHM structure is introduced by having a shorter tail domain. Thus, these functional data indicate that most of the interactions required to stabilize the motif is present. This motif is most likely close to that found on the thick filament as it is not likely that the regulated proteins on the thick filament would change the conformation of the heads but it is more likely that they would bind with only small adjustments.

The high-resolution IHM structure we present is the first for the cardiac myosin and it is the best structure to date to interpret thick filament cryoEM density maps at higher resolution. This will provide some hypotheses about how other components of the thick filament might influence the stability of and possibly the conformation of the IHM. Higher resolution Cryo-EM or Cryo-ET experiments will be needed to definitively describe the IHM structural details in cardiac thick filaments. Such structures are currently reaching ~7-10 Å resolution (Daneshparvar et al., 2020; Wang et al., 2021 & 2022) but they were done in rigor conditions in which all heads are bound to F-actin and no IHM is formed.

Some of the limitations of our interpretations of the IHM state in the context of the thick filament were already stated in Conclusions. We have added some sentences in the Discussion to clarify the limitations of our model:

“Such a variability could be linked to the differences in the regulation of each fiber. The unexpected differences in the relative orientation of the heads revealed by our IHM structure indicate that the IHM can present a unique interacting surface to the core of the thick filament from one muscle to the other (Figure 5a, 5b), likely facilitating distinct thick filament ‘inter-molecular’ interactions between crowns and with regulatory proteins such as MyBP-C and Titin. Describing these interactions awaits the high-resolution structure of the relaxed thick filament. Cryo-EM and Cryo-ET structures of the thick filaments are however currently limited in resolution^{48, 47} (Wang et al., 2022). Yet, they can benefit from high-resolution structures of IHM solved from single particle cryo-EM to fully depict how IHM form and are regulated in situ.” (l. 432-440)

The protein that was concentrated and flash frozen was thawed prior to beginning the cryo-EM experiments. Were the myosins tested for function prior to cryo-EM? Or were the myosins purified to remove dead heads? Or was the QC performed purely in the removal of some cryo-EM images?

Two quality controls on the sample were performed. The enzymatic function of the myosin construct used in this structural study (specifically single-turnover ATP assays that are thought to reflect the extent of folded-back vs open myosin heads) were tested by comparing fresh vs freeze-thawed proteins. No difference in activity was seen between the fresh protein and the previously frozen protein (and no dead-heading step was employed). Immediately after thawing, the sample was centrifugated at 15 000 g for 15 minutes to remove aggregates and each batch was analyzed by mass spectrometry (MALDI) to be sure that there was no degradation. Finally, both the images and the particles were checked and carefully selected to be sure about the quality of our map. We have edited the text in Material and Methods to fully detail the methodology used:

In the section “15-hep production and purification”, last paragraph:

“Single ATP turnover experiments from the Spudich lab (manuscript in preparation) and others⁷ confirm that 15 heptads of the proximal S2 tail allow myosin to adopt the folded-

back SRX state in solution. Freeze-thawed 15-hep myosin has been found to behave indistinguishably from freshly purified myosin in single ATP turnover experiments. SRX:DRX ratios are not sensitive to freeze-thawing.” (Supplementary material I. 242-245)

In the section “Cryo-EM data collection and processing”, first paragraph:

“The sample was thawed at 4°C and centrifugated at top speed (15 000 g) for 15 minutes before preparation. Note that for each batch, mass spectrometry analysis (MALDI) was performed after thawing in order to be sure that the proteins were not proteolyzed and that the N-terminus extensions were present. The sample was diluted at 0.3 mg/ml in 10 mM imidazole pH 7.5, 2 mM MgCl₂. 0.5 mM Mg.ATP was added and the mix was incubated 15 minutes at room temperature.” (I. 248-252)

Could the authors elaborate on the results in the context of RLC phosphorylation. Presumably the RLC phosphorylation level was 0% (maybe the reviewer has missed this, apologies if so). Native RLC phosphorylation is usually 0.4-0.5 mol Pi/mol RLC. This likely means that in native thick filaments there are 1 in every 2 heads with a light chain phosphorylation. How may this alter the IHM confirmation within a myosin dimer if the native state may approximate 1 head of each dimer is on average phosphorylated.

As detailed in the Methods section, the human RLC (MYL2) bound to the myosin heavy chain in this study was expressed in bacteria and was added to the heavy chain during the purification procedure (quantitatively replacing the endogenous mouse skeletal RLC expressed in the C2C12 cells that binds to the heterologous expressed heavy chain). As such, it is unphosphorylated – this has been confirmed by phosphoprotein staining of gel samples of the purified protein. It is known that phosphorylation destabilizes the IHM in smooth muscle myosin and that in cardiac muscle myosin, phosphorylation could increase the number of heads available for contraction (DOI: [10.1074/jbc.M113.455444](https://doi.org/10.1074/jbc.M113.455444); doi.org/10.1073/pnas.1602776113). To obtain a homogeneous sample for structural study of the IHM, we elected to purify myosin molecules containing only unphosphorylated RLC.

As we point out in Figure 3e and 4f, our structure reveals that, unlike the smooth muscle myosin IHM, the site of RLC phosphorylation in cardiac myosin is not part of the interfaces that stabilizes the IHM. In fact, the N-terminus of the RLC is free in both of the heads of our IHM structure and RLC/RLC contacts are made without implying this extension, in contrast to what has been reported for the smooth muscle myosin IHM structure.

The reviewer raises an interesting point regarding the phosphorylation state of the RLC in native thick filaments. While our structural study indicates that the RLC phosphorylation site is not present at the interface between the two RLCs, unlike Smooth muscle Myo2, it could affect the stability of the IHM interactions with the rest of the thick filament and thus influence the number of heads adopting a fully stabilized off-state on the thick filament. We are unaware of any studies addressing the phosphorylation status

of individual heads within a single myosin dimer or individual heads within a native thick filament – all such measurements have been bulk measurements that give average phosphorylation levels. Structural studies of isolated thick filaments comprised of full-length myosin with RLCs of varying phosphorylation levels would be needed to address this point. While such studies would be very interesting, they are clearly beyond the scope of this study. We thank the reviewer for this comment and we have added a statement clarifying that the RLC is unphosphorylated in this study and that future studies using RLC with different levels of phosphorylation will be necessary to determine the effect of RLC phosphorylation on the IHM stability.

Minor:

Line 256 'of the IHM in cardiac.' Should be "cardiac myosin"?

Thank you for pointing this out, it has been corrected.

The abstract is quite non-specific and doesn't really state what the findings are, given that this is outlined very neatly in sub-headings within the manuscript maybe this can be translated to the abstract to capture the headline findings more specifically.

We thank the reviewer for this comment. The abbreviated nature of the abstract (150 words) makes it difficult to spell out in significant detail our findings. With that limitation in mind, we have edited the abstract to more closely reflect the subheadings used in the body of the manuscript.

Very minor:

Some of the prose is quite dense and the sentence structure is often slightly confusing. This doesn't significantly impact the understanding of the manuscript but a little tightening would go a fair way in making this manuscript easier to read/understand.

We thank the reviewer for this comment and we have gone through the manuscript and simplified sentence structure where feasible. We hope the revised manuscript is easier to read.

Reviewer #2 (Remarks to the Author):

In this paper, Grinzato et al. use two-headed cardiac myosin II fragments expressed in, and purified from, C2C12 cells to obtain cryo-EM structures at near-atomic resolution of the interacting head motif (IHM) of myosin. This motif has the two myosin heads folded back onto themselves forming one, so called, free head (with unblocked actin binding site but low ATP turnover rate) and one blocked head where the latter also has fairly strong interactions with subfragment 2 (S2) of the two-headed myosin. The IHM is believed to be the structural counterpart of the biochemically defined super-relaxed state that exhibits 10-fold lower basal ATPase activity than free S1 in solution. Compared to earlier model studies, the work provides different structures and appreciably more detailed pictures of the interactions between the

blocked and free heads as well as between the heads and the S2 domain.

The findings are of relevance for understanding both the mechanisms for the formation of the IHM and its functional consequences. However, the results are also of interest from a translational perspective because several disease causing mutations and recently developed cardio-active drugs are believed to increase or decrease the propensity of myosin to form the IHM. In deciphering the detailed interactions behind the IHM in cardiac myosin II, the authors detect several differences from the interactions seen in other myosin II isoforms (based on earlier studies). They therefore go on to claim that their findings challenge the recent suggestion (Lee et al., PNAS 2018) that the IHM has been evolutionarily conserved in myosin II since before the origin of animals.

The paper is generally well and clearly written (but can be improved in some respects as suggested below). The motivation behind the study is well laid out in the Introduction with generally adequate citation of the literature. Likewise, I judge that the methods are sufficiently well described to allow repetition of the experiments and analyses. Presumably the Methods should be moved to the main paper.

The results provide several novel insights of critical importance for understanding the IHM and its functional importance in health and disease.

One example in physiology is how the different inter-domain interactions between cardiac and smooth muscle myosin relate to different roles of the IHM in regulation of smooth and cardiac muscle contraction. Another example is how the structure of the IHM allows the model for the cardiac thick filament to be optimized with enhanced understanding of functional peculiarities. With regard to pathologies and myosin pharmacology the results elucidate, in appreciable detail, (e.g. Supplementary Table 2 with related figures and text) the mechanism by which cardiomyopathy causing mutations and drugs might modify the IHM. Generally, the conclusions and claims made are well supported by the provided data.

Overall, I judge the study to be of high class methodologically, with well designed and carefully conducted, experiments and analyses with results that lead to a new level of understanding of the IHM in cardiac muscle. This understanding spans from individual intermolecular interactions to the filament structure. The paper addresses issues of great importance both fundamentally and translationally, e.g. for understanding of diseases and drug mechanisms, opening for new developments in therapeutics. Based on my generally favorable assessment, I strongly recommend the paper for publication. However, some issues, outlined below, require consideration.

We thank the reviewer for these very positive statements about our work.

Specific points

1. Clarity of presentation.

Whereas the presentation is rather clear overall, there are a number of issues that can be

improved, particularly to enhance the accessibility of the study to researchers outside the field of structural biology.

i. Figure 1, panels b, c

- Please increase the space between panels b and c
- Pi could be colored in the text the same way as the phosphates in the structure.
- Some key amino acid residues could be labelled to facilitate navigation

We thank the reviewer and have addressed these points. (i) The space has been increased between panels c & d in Figure 1, and the panels have been outlined with black borders to better separate them; (ii) P_i has been colored in orange in the text; (iii) some key amino acids residues have been labelled to facilitate navigation and make clear where the active site is located.

ii. Figure 3

- A line in panel b (right) from text "Car Top loop" to the structure would remove any remaining ambiguity
- Increase font size of sub-script Sm and Car for right part of panel f

These changes have been made and we thank the reviewer for improving the figure clarity with these suggestions. The font size has been homogenized and increased in panel 4e (which was formerly panel 3f). An arrow has been added from the text "Car Top Loop" to the structure.

iii. Extended Data Fig. 1.

- Too small font size in panels f and g.

The font size has been increased in panels f and g from Supplementary Fig. 1.

iv. Extended data Fig. 4

- Panel b. Would it be possible to orient the structures the same way as in panel a and also indicate the region observed in b within a box in a?

In order to remove all ambiguities regarding this figures, we added a panel showing the differences of the relative orientation of the heads in the models and in the high-resolution structure (Supplementary Figure 5c). This panel shows the heads in a different orientation compared to the panel Supplementary 5b and adds an illustration about how the interface between heads varies in the models and in the structure. We however chose to keep the panel Supplementary Figure 5b since it is important to illustrate the difference in the position of the key-residue I736 since it allows to evaluate the quality of the prediction of the head-head interfaces in the homology models.

v. Text - general

-I noted an inconsistency in use of subscript for “BH” and “FH” e.g. “BHRLC” vs “BH RLC” etc. (e.g. text on lines 195 and 199). Please check for consistency throughout.

We thank the reviewer for this remark. This has been corrected.

-Line 289: “relaxed muscle thick filament”-> “relaxed striated muscle thick filament”. Considering that smooth muscle is also a muscle.

This part is no longer in the text (comments from Reviewer#3).

-Line 303: Please define “MyBP-C” upon first use.

This was corrected, thus adding the definition of MyBP-C in the introduction: “*e.g. myosin binding protein-C or MyBP-C*” (l. 56).

-For increased clarity and readability, I suggest that the authors move Extended Data Figures 3-4 to the main paper as these figures are quite extensively discussed in the main text.

We thank the reviewer for this suggestion and we have included a comparison with model 5TBY in the main text (now **Figure 2).**

-Finally, this may not be a problem but I just remind the authors that most color blind people have problems distinguishing red and green.

We thank the reviewer for this reminder. We have tried to do our best to avoid problematic colors in the new figures.

2. Other issues

i. Lines 297-298: “increases cross-bridge kinetics” I am not sure exactly what is meant by this phrase. While I am not a native English speaker, I would prefer a more specific term such as “increases the rate of cross-bridge attachment-detachment”, “increases the rate of cross-bridge turnover” or something similar. A possible less specific alternative would be “leads to faster cross-bridge kinetics”.

We appreciate the reviewer’s comment and realize that we misquoted the data summarized in the referenced review. This part was removed in the new version of the text to fulfill the requirements of all the reviewers.

ii. Lines 75-76. The authors should consider to further weaken the statement “...challenging the claims that the IHM are conserved amongst the different myosins 2” (see also Abstract). Whereas the detailed mechanisms for the IHM seem to be different, I am not sure that the present data support the idea that IHM, as such, is not conserved amongst different myosin 2 isoforms.

We thank the reviewer for pointing this out and have clarified this statement in the paper to clearly state which particular aspects of the IHM structure may not be compared

amongst the different class-2 myosins. The introduction was rewritten in the new version of the manuscript, we thus removed this statement but discuss this point in “Cardiac and smooth muscle myosin IHMs differ”:

“Overall, the comparison between our high-resolution IHM and the only other high-resolution IHM structure solved to date provides for the first time a demonstration that major differences can be found in the structure, the stabilization and the regulation of Myo2 IHMs which might have evolved for best serve the role of this inactive state in different muscles.” (l. 376-379)”

iii. The proteins used were expressed in C2C12 cells in a highly controlled and elegant way with both the myosin heavy chains and the light chains from fast ventricular beta-myosin and all derived from the human genome. Nevertheless, the proteins are in some respect artificial by not being derived from cardiac muscle cells. This should be briefly discussed. For instance, could the use of leucine zippers to stabilize the dimerized state in any way affect the IHM structures? Would a different posttranslational modification pattern than in human cardiac cells be of relevance? I presume that this pattern is different in the C2C12 derived proteins? E.g. I presume that the regulatory light chains are not phosphorylated whereas some level of phosphorylation is generally seen in the cardiac cells. What about possible arginylation (e.g. see Rassier and Kashina, *Am J Physiol* 2019), methylation etc? Please clarify and discuss.

Since ours is the first high-resolution structure of the cardiac IHM, it is impossible to definitively state that the presence of the leucine zipper has no effect on the IHM structure as there is no high-resolution “gold standard” cardiac structure without a leucine zipper with which to compare it. However, functional studies (single ATP turnover kinetics that reflect the sequestered population of heads) show that the 15heptad construct used in this study behaves indistinguishably from longer myosin constructs including a longer human cardiac myosin construct with 25 heptads of the S2 and a leucine zipper, and from bovine heavy meromyosin (Rohde et al 2018) (which has no leucine zipper moiety), suggesting that the IHM structure of the shorter coiled coil tailed construct used in this study is unlikely to adopt a significantly different IHM structure from a longer version of myosin.

The human RLC (*MYL2*) bound to the myosin heavy chain in this study was expressed in bacteria and exchanged onto the heavy chain during the purification procedure (quantitatively replacing the endogenous mouse skeletal RLC expressed in the C2C12 cells that binds to the heterologously expressed heavy chain). As such, it is unphosphorylated – this has been confirmed by phosphoprotein staining of gel samples of the purified protein. It is known that phosphorylation destabilizes the IHM in smooth muscle myosin and that in cardiac muscle myosin, phosphorylation could increase the number of heads available for contraction (Toepfer *et al.*, 2013). To obtain a homogeneous sample for structural study of the IHM, we elected to purify myosin molecules containing only unphosphorylated RLC. As mentioned above in response to Reviewer #1, structural studies of cardiac myosin reconstituted with human RLC with different levels of

phosphorylation would be necessary to accurately assess the effect of this PTM on the IHM structure, and such experiments are beyond the scope of this study. But we can state that the mechanism by which this phosphorylation would affect the IHM stability differs from what is found in smooth muscle myosin since the N-terminus of the RLC, including the phosphorylation site, is not found in between the RLC interface and most likely will modulate interactions of the IHM with other proteins of the thick filament which may contribute to regulate the stability of the IHM. This is stated in the section “Cardiac and Smooth IHM differ”, fourth paragraph as follows:

“ The $_{car}$ IHM structure now explains this more moderate effect of RLC phosphorylation on IHM interfaces. Indeed, the RLC/RLC interface is drastically different in the cardiac IHM: the $_{BH}$ RLC extension is not part of it and the $_{FH}$ RLC N-term extension is found at the periphery of this interface even when the RLC is unphosphorylated (Fig. 4f). Thus, phosphorylation of the RLC extensions likely modulates the stability of the $_{car}$ IHM predominantly by interactions engaging other proteins in the thick filament, rather than within the IHM. This major structural difference in how the smooth and cardiac IHM form is of interest as it provides atomic detail on how post-translational modifications may distinctly regulate myosin assembly and function in striated and smooth muscles.” (l. 367-375)

With regard to other PTMs, we are not aware of any study that has systematically compared the post-translational modifications seen in C2C12-expressed cardiac myosin vs cardiac myosin isolated from human hearts. In our IHM structure, we see trimethylation of K129 and K549. Trimethylation of K129 is a well-documented PTM in different striated muscle myosins, including human beta-cardiac myosin isolated from human heart tissue (<https://doi.org/10.1021/acs.analchem.7b00113>). However, this PTM is not universally seen in either human heart tissue-derived myosin (<https://doi.org/10.7554/eLife.74919>) nor in human cardiac myosin expressed in the C2C12 system (Winkelmann and Rayment structures do not show TM-K129). Similarly, acetylation is seen by mass spectrometry in cardiac myosin purified from human heart tissue, but the number of modified residues and the position of those modified residues varies (K34-Ac, K58-Ac, K213-Ac, K429-Ac, K951-Ac, and K1195-Ac reported in <https://doi.org/10.7554/eLife.74919>; G2-Ac reported in <https://doi.org/10.1021/acs.analchem.7b00113>). Finally, the arginylation reported by Rassier and colleagues in mouse cardiac myosin (Rassier and Kashina, Am J Physiol 2019) has not been reported in cardiac myosin from human heart tissue, as far as we know. Taken together, these findings suggest that the nature and extent of these various PTMs can vary widely even among myosin isolated from a single type of tissue, making it difficult to know which PTMs (other than RLC phosphorylation) may be most relevant. However, of the cardiac myosin heavy chain residues reported to be modified in any of the above cited manuscripts (methylation, trimethylation, acetylation, arginylation), none of these residues map to areas that form head-head or head-S2 interfaces in our IHM structure.

iv. Lines 111-112 and lines 140-141. These bits of text seem to contradict each other related to priming of the lever arm. Please clarify!

“The high resolution IHM structure now reveals that the Converter position is not modified compared to what is found in an active head but the priming of the lever arm occurs only in the BH head and comes from the modification in the pliant region (Extended data Fig. 4c).”

⇒ **has been corrected as :**

“The high-resolution IHM structure now reveals that the Converter position is not modified compared to what is found in an active head but the orientation of the light chain binding region is further primed only in the BH head and comes from the modification in the pliant region (Fig. 2d, Supplementary Fig. 5d).” (l. 185-189)

v. Lines 103-104. The authors mention Movies but then refer to Extended data Fig. 1f?

The reference to Supplementary Movie 1 has been corrected as follows:

“Flexibility clearly exists in the formation of these interactions, as shown in the diagrams illustrating the flexibility of the map (Supplementary Fig. 1f).” (l. 116-117)

Reviewer #3 (Remarks to the Author):

This is an exciting paper that reports for the first time the near-atomic structure of the interacting-heads motif (IHM) for human cardiac myosin. This has been a key goal of the cardiac muscle field for several years as it reveals critical intramolecular interactions between myosin heads and the heads and the tail that create the IHM and underlie key aspects of cardiac muscle physiology (the super-relaxed state, SRX). Previous studies had approached this goal by studying IHMs from other species, but at low resolution, and by high-resolution studies of a smooth muscle IHM, which turns out not to be the same as cardiac muscle. Solution of the cardiac IHM at near-atomic resolution reveals in detail the interactions involved in the motif, the locations of some mutations that give rise to HCM and DCM, and provides a basis for interpreting and designing drugs to treat these diseases. This breakthrough will have a major impact on the field. However, the paper needs substantial work before publication. This includes clarification of many points, correcting descriptive errors, citing key papers that have led to this work that are currently ignored, improving the English, removing over-interpretation and simplifying description of the results. Fixing the points below I believe will turn this beautiful piece of work into a classic paper.

We appreciate the reviewer’s very positive statements as well as the thoughtful suggestions described below.

Major points:

1. Some key references are missing or not used appropriately. The work of Al-Khayat 2013 on cardiac thick filament structure [ref 10] is cited at many points while that of Zoghbi PNAS 2008

– which preceded Al-Khayat by 5 years – is missing. Zoghbi made the key breakthroughs in understanding vertebrate thick filament structure, which Al-Khayat then followed, and came to the same conclusions. Please properly cite this pioneering work when citing Al-Khayat. Credit should also be given to **Gonzalez-Sola, BJ 2014**. Nag 2017, on the location of HCM mutations is cited, but the parallel work of **Alamo 2017 (eLife)**, coming to similar conclusions, is omitted. Please cite appropriately. **Burgess 2007** was the first to reveal the IHM together with the course of the tail, in studies of smooth muscle myosin, and should be cited (L48).

L47 – would be helpful to add the review by **Craig/Padrón [31]** that addresses the relation between IHM and SRX.

L48: the first to show details of **nonmuscle myo2 IHM was Jung MBC 2008**, which is not cited.

L64-65, first mention of mava and OM should quote **Green Science 2016** and **Malik Science 2011**.

These references are now cited, as suggested by Reviewer 3.

L76: “SmMyo2 IHM structure would have been a good model to discuss the cardiomyopathy mutations.” Scarff 2020 and Heissler 2021 made these claims but Yang 2020 did not and should be excluded from this statement.

In order to clarify the statement about cardiomyopathy mutations, we added a sentence in the introduction:

“Based on the hypothesis that the IHM is conserved across myosin types, a recent study attempted to use homology modeling based on the $smIHM$ to predict the effects of HCM mutations on the interfaces stabilizing the off-state of human β -cardiac myosin (Scarff et al., 2021).” (l. 73-75)

L80: similar issues were discussed in Craig/Padrón [31], which should be cited.

We agree with Reviewer #3 that these references are needed and have added them. Since the initial manuscript was submitted to Nature, we initially limited the number of references to fit the format. We thank Reviewer #3 for his/her suggestions.

However, Craig/Padron could not provide information about the substantial differences in the stabilization between the two RLCs when smooth and Cardiac myosin IHM are compared, and they were also not able to detect the difference in positioning of the heads between smooth and Cardiac IHM in an accurate way, as we are reporting here. Indeed, this requires atomic resolution.

We thus cite Craig/Padron and include the reference suggested by Reviewer 3 to state that the difference in the positioning of the S2 had been previously described. This can be found in the section “Cardiac and Smooth IHM differ” in the second paragraph.

“Additionally, the interfaces stabilizing the two IHMs dramatically differ. The obvious difference at low resolution for the positioning of S2 in the tarantula IHM compared to the high-resolution of the $smIHM$ had been noticed (Yang et al., 2020; Padron et al., 2022).” (l. 288-290)

Regarding the introduction which summarizes our new findings, we have rewritten the sentence in this way:

We changed the end of introduction to emphasize this point :

“Finally, the structure reveals how the differences between the cardiac and the smooth IHM may be an adaptation to fit different muscle functions and organization.”

Is replaced by :

“Fundamental differences occur in the lever arm and the head/head interfaces of the cardiac vs the smooth IHM. These structural differences substantiate the critical role the IHM plays in providing different types of thick filament regulation in these physiologically distinct muscles.” (l. 91-93)

2. L268-307. This section (**Consequences on muscle physiology**) and related discussion should be greatly curtailed, or omitted, as it is an over-interpretation of the data. The authors imply as much in L66-68 (“The sequestered state of β -cardiac myosin has been described at 28 Å resolution¹⁰, but that is insufficient to describe precisely the IHM interfaces.”). But they then go on to do just that – fitting their high-resolution structure into the 28 Å-resolution cardiac map. Clearly, even with their beautiful high-resolution IHM, there is a lot of wiggle room when placing it in a low-resolution map, so interpreting interactions between crowns at atomic resolution is pretty much guess work.

What is needed is a high-resolution map of the filament. This would be a point worth making. But the current attempt to interpret inter-crown interactions on the basis of a 28 Å-resolution map is spurious, dilutes the high-resolution interpretation that is possible within the IHM, and could easily be misleading to the field. Another critical point in this regard is that the Zoghbi and AL-Khayat reconstructions were both based on negative stain data. A paper published earlier this year (**Koubassova BJ 2022**) shows that the IHMs in these reconstructions have collapsed roughly 35 Å radially onto the filament backbone. With such a major (artifactual) change in structure, interpreting interactions at atomic resolution is clearly fraught with danger, and would lead readers to conclude that the level of detail known is much greater than it is. The authors should omit this section of the paper along with Figs 4, 5a and ED Fig 9. See minor points below for other concerns with this section.

We fully agree with the reviewer that a high-resolution map is needed to describe atomic resolution interactions. We also agree that there is a bias in the negative staining results because of the collapse of the backbone according to the recent work of Koubassova *et al*. We thus kept a paragraph called “Consequences on thick filament regulation” (l. 381-407) but removed all the parts about fit in negative staining low-resolution maps. Now the paragraph focuses on the relative orientation of the heads between cardiac, smooth muscle and tarantula striated IHM in order to go further into the discussion of the differences.

We also cite the work of Koubassova et al., 2022 in order to evoke this bias and emphasize that fits to low-resolution negative staining should be considered with caution. In the paragraph ***“Previous homology models failed to predict the HM structure”***:

“However, maps obtained from negative staining of relaxed filaments isolated from striated muscle can be biased due to radial collapse of the heads and miscalculation of the radius of the fibers³⁹. This bias adds further uncertainty as to the accuracy of published previous fits to negative staining data derived from striated muscle.” (l. 152-155)

According to Reviewer#3’s request **Figures 4, 5a and Extended Data Figure 9 were removed from the manuscript.**

3. An important point, not discussed, is acknowledgment that the IHM solved here may not be the same throughout the thick filament. It is well accepted that IHMs in the C-zone are better ordered and have therefore been the basis of the filament reconstructions (Zoghbi, Al-Khayat). But what of IHMs in the P- and D-zones? Do the heads even form IHMs in these regions? If so, are they the same as the IHM solved here? These subtleties would be worth pointing out in the Discussion.

We thank the reviewer for this important point and we have added to the discussion the need for cryo-ET to begin to resolve this issue:

*“Such a variability could be linked to the differences in the regulation of each fiber. The unexpected differences in the relative orientation of the heads revealed by our IHM structure indicate that the IHM can present a unique interacting surface to the core of the thick filament from one muscle to the other (**Figure 5a, 5b**), likely facilitating distinct thick filament ‘inter-molecular’ interactions between crowns and with regulatory proteins such as MyBP-C and Titin. Describing these interactions awaits the high-resolution structure of the relaxed thick filament. Cryo-EM and Cryo-ET structures of the thick filaments are however currently limited in resolution⁴⁸ (Wang et al., 2022). Yet, they can benefit from high-resolution structures of IHM solved from single particle cryo-EM to fully depict how IHM form and are regulated in situ.” (l. 432-440)*

4. Fig 4. There are a number of minor points that would need to be corrected in the figure and legend. But, as stated, this figure should anyway be omitted. Likewise Fig. 5a and ED Fig 9, as they imply a resolution of this interface that has not been achieved.

These figures were omitted.

5. ED Fig 1. The image quality/resolution needs to be much better. The figure quickly becomes pixelated when enlarged. 2D classes are too small and no secondary structure is visible. “(e) Shows the flexibility within the different classes used to reconstitute the map.”: (e) should be (f), and “local flexibility” should be explained. The legend says two orientations, but 3 are

shown. What are those orientations – explain in relation to the 3D map? “(f) plots the different orientations of the particles.”: “f” should be “g”.

Supplementary Figure 1 was edited in order to increase the readability. The figure legend has also been changed.

6. L106. Please show representative examples of cryo-EM densities of the interaction sites of the side chains to give the reader a clear idea of how well the predicted interactions are actually seen in the density map. Looking at the fit of the PDB model to the map seems to show substantial regions where there is empty density and also density outside the map – can the fitting be improved? An example is the fitting of ADP.Pi to the map, which does not look unique.

The EM density is clear about describing the presence (and location) of the ADP and Pi, but at ~3.2 Å resolution, it is not possible to describe exact coordination and thus provide precise side chain position. However our map is of good quality for a 3.2 Å resolution structure. We have now added more convincing figures (Supplementary Figure 3), there is no doubt of the presence and location of Pi in both of the heads. The cross correlation coefficients for the nucleotides are 0.77 and 0.89 for the BH and the FH respectively. The fact that the main chain of the protein can be clearly defined in density describes the conformational state that the heads adopt and also restricts the possibilities for side chain orientation, in a way that would not be possible for maps at 4.5 or higher resolution. Overall, this map and this resolution is thus of really high quality and there is no region that is built outside the map.

7. L84. The construct is stated to be native, which suggests coming from tissue. It is recombinant.

We thank the reviewer for pointing this out; we have removed the term “native”.

8. Methods (ED p11). The construct is stated to be human, but involves replacement of mouse RLC with human RLC. Some quality control would be useful to show: how much was exchanged, how much was the mouse RLC depleted? It would help to provide a cartoon of the vector map with the construct used to express the recombinant protein. Is the FLAG tag only attached to the ELC, not the HC? Not clear. It would also help to have a figure showing the ion-exchange profile to give an idea of the salt concentration for the protein elution.

Exchange of the human RLC onto the β -cardiac myosin heavy chain, replacing the endogenous mouse skeletal RLC, appears to be at least 95% complete as no residual mouse RLC is visualized by SDS-PAGE of the purified protein. We add a representative gel of the 15hep myosin before and after RLC exchange in the figure Fig. R#0 below. As detailed in the methods section, the FLAG tag is only on the N-terminus of the ELC, while the exchanged human RLC has an N-terminal 6xHis tag. Both are removed via TEV-cleavage after RLC exchange has taken place. The heavy chain has only an 8 amino acid

affinity clamp peptide at the C-terminus for surface attachment to a PDZ protein for motility studies. The 15hep protein typically elutes at ~ 200-250 mM KCl.

1. FLAG resin post binding to WT 15hep lysate
2. FLAG resin post mouse RLC depletion
3. FLAG resin post human RLC exchange
4. purified WT 15hep

Figure R#0 : SDS-PAGE showing the 15-hep before and after RLC exchange.

In the section “15-hep production and purification”, we added a quality control that we have done to be sure that our constructs were still able to form the SRX last paragraph:

“Single ATP turnover experiments from the Spudich lab (manuscript in preparation) and others⁷ confirm that 15 heptads of the proximal S2 tail allow myosin to adopt the folded-back SRX state in solution. Freeze-thawed 15-hep myosin has been found to behave indistinguishably from freshly purified myosin in single ATP turnover experiments. SRX:DRX ratios are not sensitive to freeze-thawing.” (l. 242-245 in Supplementary material)

In the section “Cryo-EM data collection and processing”, we added that we analyzed by mass spectrometry analysis each batch in order to be sure that the proteins (especially the light chains) were not proteolyzed. first paragraph:

“The sample was thawed at 4°C and centrifugated at top speed (15 000 g) for 15 minutes before preparation. Note that for each batch, mass spectrometry analysis (MALDI) was performed after thawing in order to be sure that the proteins were not proteolyzed and that the N-terminus extensions were present. The sample was diluted at 0.3 mg/ml in 10 mM imidazole pH 7.5, 2 mM MgCl₂. 0.5 mM Mg.ATP was added and the mix was incubated 15 minutes at room temperature.” (l. 248-252 in Supplementary material)

For homology modeling, what method/program was used? Reference? “While some ambiguity exists at times in these interfaces, the electron density is good for the main chain and the large hydrophobic side-chains”. That means some side chains reported are based on simulated models, not side chain density? Please clarify.

We added some precision about this procedure in the Material and Methods section. The text is now as follows:

“The first coordinates for the IQ2/RLC region was obtained by homology modeling with the software Swiss-Model (Waterhouse et al., 2018) using the SmMyo2 IHM¹ (PDB code 7MF3) as a template.” (l. 272-274 in Supplementary material)

The additional step of minimization with molecular dynamics simulation during refinement is a calculation procedure that we already used in other published work. The principle is to minimize the position of the side chains at the interfaces in order to compute realistic interfaces. Indeed, there is always final step of refinement (real space or reciprocal space) after this step in order to fit into the map. When we used this method in our last cryo-EM work (Robert-Paganin et al., 2021) on several actomyosin complexes at medium to high-resolution, the addition of the molecular dynamics simulation step improved both the quality of the model and the correlation coefficient between the model and the map. Moreover, it allows to compute realistic interfaces, specifically in the regions where the long sidechains (e.g. R or K) are not fully defined in density. We added some text to clarify this point:

“A final run of real space refinement was performed against map 1. This method, combining steps of molecular dynamics minimization at the interfaces of complexes and final steps of refinement allows to compute realistic interactions for the side chains that may not be fully defined in density. The strength of this calculation routine is to avoid bias due to molecular dynamics simulations, which are independent to refinement in the map and focused on the interfaces. The methodology was previously used successfully in the refinement of cryo-EM maps (Robert-Paganin et al., 2021). These steps of molecular dynamics simulations improved the correlation coefficient between the model and the map (Robert-Paganin et al., 2021).” (l. 282-288 in Supplementary material)

Why was “bovine β -cardiac myosin PPS complexed to Omecamtiv mecarbil2 (PDB code 6N69)” used for model building instead of 5n6a (bovine) or 4p7h (human cardiac) motor domain? Note that, anyway, it should be 5n69, not 6n69 (Fig 5 legend, Model building, and ED Table 1).

We used the 5n69 structure because it is the highest resolution structure available for cardiac myosin in the pre-powerstroke state and it includes IQ1/ELC. The 4p7h structure is not a pre-powerstroke structure, it does not correspond to a myosin structure with a primed lever arm. We thank the reviewer for pointing out the typo, which we have corrected.

9. L124. The correct ref for 5tby is not [27], but Alamo eLife 2017. One could question whether it is more appropriate to use MS03 in the comparison (not validated and not in the PDB database) rather than 5tby, which is. 5tby should be added to the comparison in ED Fig. 4a as it is the only deposited human cardiac IHM PDB. A confusing point: L142-143 states that in MS03

the heads would be in a pre-pre-stroke conformation. They also say that 5tby and MS03 are similar enough to study only one (MS03) (L124-125). And yet according to Alamo 2017 eLife, 5tby was thought to be in a pre-stroke conformation (not pre-pre). This appears to be self-contradictory, and further suggests that 5tby should also be compared.

We had initially chosen to present only MS03 since the models were discussed on the following criteria: (i) head-head interface, (ii) presence of a kink in the BH lever arm; (iii) position of the coiled-coil. Both MS03 and 5TBY display similar features on the chosen criteria. However, we agree with Reviewer #3 that given the predictions on HCM mutations provided by the paper of Alamo et al., 2017, it is appropriate to include the model in the comparison. Former **Extended data Fig. 4** is now divided into two figures (**Figure 2 & Supplementary Figure 5**). Since Alamo et al. have done the prediction of the mutations, we include the comparison of our structure with 5TBY in the main text. A specific paragraph has been included in order to compare the structure with the previous models, as proposed by Reviewer #2: see **“Previous homology models failed to predict the IHM structure”** (l 147-195).

To answer the Reviewer’s point on the differences between the models 5TBY and MS03, we are adding below figures to illustrate how similar the two models are, and equally wrong in predicting the conformation of the BH, in the position of the Converter/Lever arm. While *Alamo et al. 2017* had not distinguished the difference in their model from what was previously called PPS from crystal structures, Spudich et al discussed the difference in the BH conformation models compared to the pre-powerstroke structures and thus called the conformation seen in either 5TBY or MS03 pre-prestroke conformations. In fact, the same conformation was also proposed for the FH. Our high-resolution structure now shows that this interpretation of low resolution map is wrong for both models. In reality, both heads do adopt a pre-powerstroke conformation rather than a novel pre-prestroke conformation, as the hinge in the pliant region is the key element to position the lever arm of the BH in a more primed orientation. These comparisons indicate that the models are equally wrong in providing the atomic structure of either the BH or the FH heads, and thus in predicting the interactions between heads. As an example, the important feature incorrectly modelled corresponds to the position of the Converter and the conformation of the Top loop which drastically affect prediction for the stabilizing interactions in the IHM as the FH Converter (and in particular the Top-loop) is part of the head/head interface (as shown in **Figure R#1a, R#1b, Fig. 2 and Supplementary Figure 5**).

Figure R#1a : Conformation of the BH heads : A-C and E : Comparison of the BH head conformations in our IHM structure and in the 5TBY and MS03 models. D: The BH-head adopts a pre-powerstroke conformation, as previously crystallized for the ADP.Pi state of cardiac myosin. The priming of the lever arm is linked to a drastic kink in the pliant region, not in a change in the Relay or Converter position as previously proposed from low resolution models (5TBY / MS03).

Figure R#1b : Conformation of the FH heads : A-D : Comparison of the FH head conformations in our IHM structure and in the 5TBY and MS03 models. E: The FH-head and the BH-head in our high-resolution IHM structure adopt a pre-powerstroke conformation, as previously crystallized for the ADP.P_i state of Cardiac myosin. F: both the MS03 and 5TBY proposed a similar orientation of the Converter. This proposition was not

correct as it drastically differs from that of the high-resolution IHM structure and thus wrongly position the Top loop which is a primary site of head/head interaction of the IHM.

10. L149-213. I found this an extremely detailed and complex description. It would be very helpful if it could be simplified/reduced to make it more accessible to the reader. Likewise ED Table 2. As it is, this section will not be very clear or accessible to a non-specialist reader.

We have changed the text to describe it more clearly. We now see that the first paragraph might have been difficult to read. For the description of the interactions we tried to simplify as much as possible, without omitting important information. The new version of the section “Interactions stabilizing the IHM” is presented I. XX-XX.

The Supplementary Table 2 details the residues involved in the different interfaces in the columns on the left and compares with the residues involved in the smooth IHM to show how they differ. The five interfaces are listed, with subdivisions for the Head/Head interface so that the reader can find which residues correspond to the different names of the myosin head. We admit that this table is meant for readers interested in the details of these interfaces, but we think that in fact the fact that the name of the regions is provided with the residues involved will help these readers interested in the detail of the interactions. We have however added the following prior to the table to help readers to understand it.

“This table details the residues involved in the different interfaces of the cardiac IHM in the columns on the left and compares with the residues found in similar contacts for stabilization of the smooth IHM on the left. In particular, an orange background allows to find which interactions are drastically different between the smooth and the cardiac IHMs. The five interfaces are listed one after the other, with subdivisions for the Head/Head interface so that the reader can easily find which residues correspond to the different parts of the myosin head involved in these interactions.” (I. 313-318 in Supplementary material)

11. L218. The refs here should be [11] and [12], not [31]. Also Yang [14] pointed out a major difference between the filament IHM and that in SmMyo2, in that S2 is shifted by 20 Å compared with the IHM in the thick filament (and replaced by seg3). This is a major difference, making it clear that IHMs are not all alike, when seen in detail. It seems that the authors may be aware of this as they state (L62-64): “Interestingly, the recent SmMyo2 IHM structures challenges the Mesa hypothesis¹⁴ in that these structures are significantly different from the human β-cardiac homology models²⁰.” Yang [14] noted this early on and should be acknowledged on L218.

We changed the text as suggested by the reviewer:

“Additionally, the interfaces stabilizing the two IHMs dramatically differ. The obvious difference at low resolution for the positioning of S2 in the tarantula IHM compared to the high-resolution of the smIHM had been noticed (Yang et al., 2020; Padron et al., 2022). Yet, the fact that differences would be found in the head/head and RLC/RLC interfaces was totally unexpected

since these interfaces were presumed to be conserved in all class-2 myosins due to lack of high-resolution information^{16,17,28,11,12,13}.” (l. 288-292)

Minor points:

1. Most of the paper is pretty well written, but **there are lots of non-idiomatic English** expressions. Please have one of the native English-speaking authors correct these to make the work as clear as possible to readers.

We thank the reviewer for pointing this out and have endeavored to clarify the manuscript text as much as possible.

2. L25: interacting-heads motif (hyphenated, heads plural).

This has been corrected.

3. L49. Worms should be flatworms (to distinguish from earthworms).

This has been corrected.

4. L51. Certainly, the isolated motif of interacting heads has been conserved [refs 11, 12] as well as in muscle thick filaments [ref 27], even if the details vary as shown in the current paper, and previously [ref 14].

We agree with Reviewer #3 that indeed the IHM would appear conserved at low resolution since it consists in two folded back heads (FH and BH) with extra interactions with a coiled-coil. And this Myo2 double-headed motif is key for the regulation of the myosin head and prevent them from being active.

However, the high-resolution structure of Cardiac Myo2 and its comparison to Smooth Myo2 reveal key differences between the two myosins: (i) relative orientation of the heads and head-head interface; (ii) position of the coiled-coil; (iii) hinges of flexibility (kink angles of the lever arms); (iv) conformation of the phosphorylatable RLC N-term extension.

Previous models (5TBY, MS03, MA1) were not able to highlight how the cardiac IHM forms and differs from that of Smooth IHM which is the only other IHM whose structure is currently known at high-resolution (Fig. R#1a, R#1b, 2, 3, Supplementary Movie 4).

These differences cannot be considered as details since they result in (i) substantially different interfaces in the two myosins and (ii) different consequences of the regulatory phosphorylation in the RLC N-term. In the new version of the manuscript, the consequences of these differences are discussed in the paragraph “Consequences on thick filament regulation” (l. 381-407). The fact that such a variability could not be anticipated is one of the major novelties of this work and illustrates that high-resolution structures are needed to properly analyze and compare the off-states of myosins, and subtle differences that allow them to be regulated distinctly depending on the muscle.

“From the low resolution examined (>2 nm), all types of muscle myo2 IHM look alike and this motif has been assumed to be conserved¹¹”

has been changed as follows :

“From the low-resolution examined (>2 nm), all types of muscle myo2 IHM look alike and the interactions within this motif have been assumed to be conserved¹⁶” (I 46-48).

5. L85-86. I found this description of the section of coiled-coil (residues 1410-1625) slightly confusing as it is just a middle stretch of tail, not the entire length of the tail beyond S2. One has to look carefully at the SmMyo2 papers to see which bit of the tail they mean. Maybe indicate the region on ED Fig 2, or omit ED Fig 2 as it is not really necessary, having been shown in the previous SmMyo2 papers.

We changed Supplementary Fig 2 to make this clearer. We refer in the legend to the previous papers but kept this ED figure as it helps readers to visualize the difference between smooth and cardiac IHMs.

6. L90-91. “Only a portion...” Is this based on 3D classification? If so, what other structures were found?

The other classes found in 3D classification correspond to HMM that do not form the IHM. In this case, the S1 heads are not involved in interactions with S2 or another head. Due to the high flexibility that an ‘on’ state of an HMM can undergo, we could not rebuild a map for the double headed ‘on’ state, but could average the particle on the motor domain. A map could be built for single-heads in the pre-powerstroke (PPS) state gathering most of the particles that had not been selected for the structure of the IHM. The resolution obtained was high (<2.7Å resolution), and did not reveal any density that could correspond to the S2 docking site or any other region bound to the motor domain. This class was thus not really part of the IHM story. Addressing what it represents would require further experiments that fall outside the scope of this work.

7. L99: How well does the post-hydrolysis model correspond with the map? What is the strong density near phosphate 1 of ADP in the FH? The adenine portion does not follow the density well and there is unmodelled density near the benzene ring in the BH. Overall, the fitting shown in Fig. 1c-e (and in the fitted 3D maps) does not look so impressive. Can it be improved?

In order to better illustrate the fit of the active site of each head in the cryo-EM map, a supplementary figure was added (Supplementary Fig. 3). In Supplementary Fig. 3a, 3b, the fit of the ADP.P_i + Mg²⁺ ion in the density is obvious. The correlation coefficient between the nucleotide+Mg²⁺ and the cryo-EM map is 0.89 and 0.77 in the FH and in the BH respectively (Supplementary Fig. 3a, 3b). The previous figure was likely a bad choice of orientation. In Fig. 1, we show all the sidechains around the nucleotide. According to the requirements of Reviewer #3, we also added a figure showing the conformation of the connectors and specifically the Switches-1 and -2, thus illustrating the closure of the

back door in $_{\text{Car}}\text{IHM}$ (Supplementary Fig. 3c, 3d) and in SmIHM (Supplementary Fig. 3e, 3f). These figures show without ambiguity that the active site is in a pre-powerstroke conformation in $_{\text{Car}}\text{IHM}$ with a cryo-EM map compatible with a closed back door.

A detailed comparison of the conformations of the heads in $_{\text{Car}}\text{IHM}$ and SmIHM is also shown in Supplementary Fig. 9.

8. L103-104. Movies of 2D classes are mentioned but not shown. ED Fig. 1f and 1g needs to be properly explained.

The sentence is now as follow:

“Flexibility clearly exists in the formation of these interactions, as shown in the diagrams illustrating the flexibility of the map (Supplementary Fig. 1f).” (l 116-117).

The legend of Figure 1f and 1g was also modified as follows:

*“**Supplementary Figure 1 – Cryo-EM map and validation.** (a) Shows the steps of processing. A representative micrograph on gold grids, 2D classes, and the steps of refinement and masking. (b) and (c) show the Fourier Shell Correlation and the overall resolution on Map 1 and 2 respectively (criterion 0.143). (d) and (e) display the local resolution for each map with two orientations. (g) Shows the flexibility of the density map shown in d. (f) plots the angular particle distribution.” (l. 30-34 in Supplementary material)*

9. L106. Should be molecular dynamics simulations

This has been corrected.

10. L107. Please define the ELC and RLC N-terminal extensions. Which residues? References for each?

A sentence was added in Methods to define the N-terminal extensions of the light-chains and what could be clearly seen in our IHM structure:

“Finally, two essential regions of regulation, the N-terminal extensions (N-term extensions) of the ELC and RLC were partially rebuilt in density (Fig. 1f, see Methods).” (l. 120-122)

In Methods:

“The ELC and the RLC N-terminal extensions range from residues 1-48 and 1-27 respectively, and density allowed us to attribute the conformation of the following residues : BH-ELC: 39-195 ; BH-RLC: 20-162 ; FH-ELC: 39-195 ; FH-RLC: 20-163.” (l. 289-291 in Supplementary material)

11. L108-109. Both heads have similar stability. This is surprising based on previous studies. Could it be because the 200,000 molecules in the reconstruction were specifically included because they gave a good match to the fully formed IHM (with BH and FH in their correct positions)? What maps did the 300,000 unused particles generate? For example, did any of

those maps show a blocked head but not a free head (disordered?), as one might have expected? If so, this could support the concept of FH mobility. See also Minor point #6 above.

As described above in point #6, the rest of the particles correspond to single heads or disordered HMM (active state). The high-resolution map generated did not allow us to see any density around the motor domain that may have been interpreted as being part of a S2 region. In addition, the lever arm were disordered indicating that we were not able to identify a class with only one head bound to the S2 coiled-coil.

12. L112-113. This (pre-powerstroke state) appears to differ from Heissler for SmMyo2, who proposed a different state, off the main crossbridge cycle. This could be an interesting discussion point, given that smooth myosin is switched off 10X more than cardiac, which Heissler explains by a different, more inhibited, structure from a classical pre-powerstroke state. Is RMSD based on the C- α atom or include the side chains?

We had noticed this difference but didn't report it previously. It is a true additional difference between the smooth and cardiac IHMs. We have now added a paragraph to discuss the differences in states between cardiac and smooth muscle myosin IHMs:

“Importantly, the high-resolution structures of $carIHM$ and $smIHM$ strongly diverge (Supplementary Fig. 9a, 9b). First, the $smBH$ and $smFH$ motor domains are not in a canonical pre-powerstroke conformation (Heissler et al., 2020), unlike $carBH$ and $carFH$. The conformation of the P_i release tunnel (Llinas et al., 2015) differs: in $carIHM$, but not in $smIHM$, the tunnel is closed by a conserved salt bridge formed between the switch-1 arginine (R243 in $carIHM$ and R247 in $smIHM$) and switch-2 glutamate (E466 in $carIHM$ and E470 in $smIHM$). This closed backdoor blocks the escape of the phosphate in the BH and the FH of $carIHM$ (Supplementary Fig. 9c, 9d). In contrast, the positions of switch-1 and switch-2 results in an open backdoor for both heads in $smIHM$ and the critical salt bridge cannot form (Supplementary Fig. 9c, 9d). Finally, the priming of the lever arm differs in the BHs of the two IHMs. It is 10° more primed in $carIHM$ (Supplementary Fig. 9e), but is more similar in the FHs (Supplementary Fig. 9f).” (l. 278-287)

We also added a sentence in discussions to state that these differences between the IHMs highlights the differences in regulation of different muscles:

“(ii) smooth and cardiac IHMs are drastically different and (iii) these structural differences highlight the molecular basis for the unique physiology and regulation of these different muscles. In addition, comparison with the tarantula striated muscle IHM exemplifies how distinct interfaces may occur amongst different striated muscles resulting in a large diversity of the stability of the IHM depending on the species and the evolution of the muscle.” (l. 421-423)

13. L115-116. The larger kink was seen in all 3 SmMyo2 structures solved, also in past Alamo papers and in Wendt 2001. Reference to these would be appropriate. Same for L187-188.

While the difference between the BH and FH lever arm orientation was recognized in all the papers cited by the reviewer, the fact that it involves the pliant region was not obvious and is mentioned in this study. It needed high resolution to be defined correctly. Therefore we do not cite these previous studies, the sentence is meant to precisely explain where the kink occurs.

We provide additional supporting data: **Supplementary Movie 4, Figure 2 and Supplementary Fig. 5** to better illustrate how the structure differs from previous models. Unlike previous studies, we are now able to provide a detailed understanding of the hinges required for the IHM to form and that matter for its stability.

We have not changed this particular sentence as it precisely describe the structure:

*“The major and only difference is the larger kink in the pliant region³⁶ of the BH lever arm. This larger kink between the Converter and the lever arm alters the ELC/Converter interface (**Supplementary Fig. 4d, 4e, 4f**) and the so-called musical chairs (**Supplementary Fig. 4e**), a network of labile electrostatic interactions involved in the control of the lever arm dynamics at this interface²⁷. ”*

We do mention the previous work this way :

“The role of flexible hinges in the lever arm for the formation of the IHM had been anticipated³⁰(Wendt et al., 2001, Alamo et al., 2008) but they had never been described precisely.” (l. 244-245)

14. L145-147. A comparison with Heissler SmMyo2 structure would be informative here.

The comparison of the motor domain conformation for the two IHMs was added as a part of the paragraph “Cardiac and Smooth muscle myosin IHM differ”:

*“Importantly, the high-resolution structures of *car*IHM and *sm*IHM strongly diverge (**Supplementary Fig. 9a, 9b**). First, the *sm*BH and *sm*FH motor domains are not in a canonical pre-powerstroke conformation (Heissler et al., 2020), unlike *car*BH and *car*FH. The conformation of the P_i release tunnel (Llinas et al., 2015) differs: in *car*IHM, but not in *sm*IHM, the tunnel is closed by a conserved salt bridge formed between the switch-1 arginine (R243 in *car*IHM and R247 in *sm*IHM) and switch-2 glutamate (E466 in *car*IHM and E470 in *sm*IHM). This closed backdoor blocks the escape of the phosphate in the BH and the FH of *car*IHM (**Supplementary Fig. 9c, 9d**). In contrast, the positions of switch-1 and switch-2 results in an open backdoor for both heads in *sm*IHM and the critical salt bridge cannot form (**Supplementary Fig. 9c, 9d**). Finally, the priming of the lever arm differs in the BHs of the two IHMs. It is 10° more primed in *car*IHM (**Supplementary Fig. 9e**), but is more similar in the FHs (**Supplementary Fig. 9f**).” (l. 278-287)*

15. L166. Ref for PHHIS?

The reference was added: Spudich, Pflugger, 2019.

16. L203-204. Their relative position is distant – not completely clear. Is the meaning that the patches are at a distance from each other?

Yes, this is what we meant: that the different parts of the IHM involved in interaction are distant. We changed the sentence to make it clear.

*“Interestingly, all these interfaces in *car*IHM are quite modest in surface area and involve several electrostatic interactions found at a distance from one another (Supplementary Table 2).” (l. 262-263)*

17. L205-206. Most of the contacts made are dynamic in nature. Evidence, reference?

We apologize if the sentence was not clear. The evidence comes from the fact that at the interface, the side-chain densities is sometimes difficult to visualize and necessitates higher contour. To clarify, we modified the sentence which is now as follows:

*“In addition, most of the contacts stabilizing the IHM interface are dynamic in nature. Electrostatic interactions between long side chain residues can indeed be established with more than one partner. The concept of ‘musical chairs’²¹ has been proposed to describe such interfaces made of dynamic polar contacts, as described for the Converter/ELC interface²¹. Indeed, molecular dynamics has shown that electrostatic interactions at the Converter/ELC interface are labile and can arrange differently, thus maintaining a certain flexibility of the lever arm²¹. The structure demonstrates that ‘musical chairs’²¹ are found at different interfaces and hinges of flexibility of the *car*IHM. Such a property of these interfaces is key for allowing prompt regulation.” (l. 263-271)*

18. L208. It would help if the authors provided a short intuitive description of what is meant by “musical chairs”.

We added a sentence and modified the text to make a definition of musical chairs as stated in the response minor comment of this reviewer 17.

19. L231-233. This is no surprise, given the well-established similarities between smooth/nonmuscle which are different from the similar cardiac/skeletal.

The current structure is the first that provides strong evidence regarding the differences in the sequestered states between smooth/non-muscle/tarantula and cardiac/skeletal muscle, and this statement may not be obvious to most readers. However, to satisfy this comment, we removed this statement from the current version of the manuscript.

20. L237-240. These facts (“Surprisingly, the cardiac S2...”) were clearly anticipated by Yang [14], which should be referenced. Their ED Fig 6 is similar to Fig 3d here. Scarff and Heissler did not discuss this.

The sentence was changed in order to credit Yang and colleagues for their observation that the first fragment of the S2 coiled-coil was different in tarantula skeletal muscle (3JBH) compared to that found in the smooth IHM, based on low resolution EM maps of thick filament.

“Surprisingly, the cardiac S2 interacts extensively with the Mesa of the BH, in drastic contrast to the position S2 adopts in the SmMyo2 IHM, which only allows for a few interactions with the FHLoop2. In fact, it is Seg3 that interacts with the BH Mesa in SmMyo2 IHM.”

has been changed as follows, crediting the recent review written by R. Craig and R. Padrón:

*“Additionally, the interfaces stabilizing the two IHMs dramatically differ. The obvious difference at low resolution for the positioning of S2 in the tarantula IHM compared to the high-resolution of the *sm*IHM had been noticed (Yang et al., 2020; Padron et al., 2022).” (l. 288-290)*

21. L245-246. It’s not clear from Fig 3e that S19 and S15 positions are conserved. Only S19 is shown.

The sequence alignment shows how the serine is conserved. Despite its conservation in its position in the sequence, it is not conserved in the interaction it makes between the RLCs. To make this clearer, we changed Fig. 4e in order to indicate that both serines can be phosphorylated and are conserved in when they are positioned in the sequence compared to the first helix of the RLC.

In the legend, *“The phosphorylation is represented in red with “P”.”*

has been changed as follows : *“The phosphorylation sites (S19 for SmRLC, S15 for CarRLC) are indicated as a red serine “S”.” (l. 334-335)*

The text was also changed to make this important point as clear as possible :

*“Both CarRLC and SmRLC are phosphorylatable in a charged and disordered N-terminal extension (N-term extension) which is not conserved in sequence (Fig. 3e), although from the globular N-terminal lobe of the RLC, the position of the two phosphorylatable serines S19 in SmRLC and S15 in CarRLC is conserved (Fig. 3e). In SmMyo2, RLC phosphorylation acts as an on/off switch to disrupt the SRX and activate the muscle³². This was explained in the Smooth IHM structures^{13,15} where the *BH*RLC N-term extension interacts with seg3 and the *FH*RLC N-term extension is directly part of the RLC/RLC interface. S19 Phosphorylation thus disrupts these IHM interfaces, switching on the motor (Fig. 3e). In cardiac myosin, RLC phosphorylation is not strictly necessary to activate cardiac muscle but modulates its activity³³). The cardiac IHM explains this more moderate effect on IHM interfaces, beyond the fact that seg3 is not part of stabilizing the cardiac IHM. The RLC/RLC interface is drastically different in the cardiac IHM: the *BH*RLC extension is not part of it and the *FH*RLC N-*

term extension is only found at the periphery of this interface (Fig. 3e). No density allows precise positioning of the $_{FH}RLC$ S15 as it is not a major part of the RLC/RLC interactions. Thus, phosphorylation would only modulate the stability of the IHM in cardiac myosin. This major structural difference in how the smooth and cardiac IHM form underlies the distinct regulation in striated and smooth muscles.”

has been changed as follows :

“Both $_{Car}RLC$ and $_{sm}RLC$ are phosphorylatable in a charged and disordered N-terminal extension (N-term extension) which is not strictly conserved in sequence, although from the globular N-terminal lobe of the RLC, the position of the two phosphorylatable serines, S19 in $_{sm}RLC$ and S15 in $_{Car}RLC$, is conserved (Fig. 4f). In $_{sm}Myo2$, the phosphorylation of RLC S19 acts as an on/off switch to disrupt the IHM and activate the muscle³². This was explained in the smooth IHM structures^{13,15}, where the $_{BH}RLC$ N-term extension interacts with Seg3 and the $_{FH}RLC$ N-term extension is directly part of the RLC/RLC interface. S19 phosphorylation thus disrupts both of these $_{sm}IHM$ interfaces, efficiently switching on the motor (Fig. 4f). In cardiac myosin, RLC S15 phosphorylation is not strictly necessary to activate cardiac muscle but modulates its activity³³. The $_{Car}IHM$ structure now explains this more moderate effect of RLC phosphorylation on IHM interfaces. Indeed, the RLC/RLC interface is drastically different in the cardiac IHM: the $_{BH}RLC$ extension is not part of it and the $_{FH}RLC$ N-term extension is found at the periphery of this interface even when the RLC is unphosphorylated (Fig. 4f). Thus, phosphorylation of the RLC would not drastically affect the RLC/RLC interface and likely modulates the stability of the $_{Car}IHM$ predominantly by interactions engaging other proteins in the thick filament. This major structural difference in how the smooth and cardiac IHM form is of interest as it provides atomic detail on how post-translational modifications may distinctly regulate myosin assembly and function in striated and smooth muscles.” (l. 359-375)

22. L257. “Underlies” sounds like a statement of fact. But it is an interpretation.

This was changed:

“This major structural difference in how the smooth and cardiac IHM form is of interest as it provides atomic detail on how post-translational modifications may distinctly regulate myosin assembly and function in striated and smooth muscles.” (l. 372-375)

23. L277. Refs 35-37 are inappropriate. Huxley and Brown [35] discovered the perturbation, but did not know anything about it structurally. Luther [36] doesn't address the issue of interaction between crowns. And Al-Khayat [37] is an early and inadequate reconstruction to answer this question. The two appropriate references to quote would be Zoghbi 2008 and Al-Khayat [10]. However, this is part of the section that we suggest should be deleted anyway (Major point #2).

This has been corrected in the new version. These parts of the text were removed in the new version according to request in the major points.

24. L294-295. The inter-crown interaction in tarantula does involve the RLC. It appears not to in Fig. 4d because the authors have placed the cardiac IHM into the tarantula map. But that is inappropriate. When the tarantula IHM is placed in the tarantula map, there is interaction involving the RLCs. This is because the tarantula RLC has a much longer N-terminal extension than cardiac (Alamo 2008, Brito 2011, Alamo 2016). This incorrect reasoning (and several mistakes in the legend of Fig 4) is another reason that this figure and the section Consequences on muscle physiology should be removed.

We have removed all the references to inter-crown interactions in the new version of the manuscript. The current section named “Consequences for thick filament regulation” (l. 381-407) focuses on how the conformational differences of the isolated IHMs may have some consequences in the exposed surfaces when these motifs are incorporated to a filament.

25. L313. The authors quote refs 21, 27, 28 and 40. They omit reference to Alamo eLife, a parallel study to Nag 2017 [28]. Instead they refer to Alamo Biophys Rev 2017, a review of the filament IHM—which should be quoted in the IHM introductory section. I am guessing this is simply a mix-up of references. The eLife paper should be quoted, but is currently not in the reference list at all. The same on L315. Similarly, L322-323. Quote Nag 2017 and Alamo 2017 (eLife) who first suggested this.

We apologize for the omission of Alamo Biophys Rev 2017 which was a mistake. We indeed wanted to cite this work but it was a mistake when using the formatting for bibliography. This is now corrected in the new version of the manuscript where these data (Alamo et al., 2017 and Nag, 2017) is cited.

Since we had to shorten the manuscript this paragraph on mutations has been removed but these references appear now as refs 38 and 26.

26. L337-339. As stated earlier, reference to inter-crown interactions should be toned down or removed (major point 2 above). L338. Fig 5a is also over interpreted for the same reason and should be omitted. L342-343: Again premature interpretation of inter-crown interface, when we really do not know its structure at near atomic resolution. Similar for L345-349.

This has been corrected in the new version. These interpretations of inter-crown interfaces were omitted.

27. L345 and Fig 5a. These DCM mutations and the figure and legend are very confusing. Just one example – E525 is implied to be in the inter-crown interface (but doesn't appear to be so at all) and as destabilizing the IHM (L582-583). The reasoning and statements have little support. In fact E525K has been shown to stabilize, not destabilize SRX (Ras Ricci BioRxiv 2022), and the mechanism appears to have nothing to do with the inter-crown interface, but in fact to stabilization of the BH/S2 intramolecular interface. Again, the inter-crown section should be deleted. One might add that the mutational analysis in general should be improved, as some of the mutations discussed are not known to be pathogenic and therefore unlikely to be

structurally significant (cf. Alamo eLife, which analyzed only pathogenic or likely pathogenic mutations).

Due to space limitations and in order to make the essential messages of this work clear, we removed the detailed analysis of the mutations. Thus the paragraph “Inherited cardiomyopathies and therapies”, previous Extended Data 1 and associated figures were removed. We added a specific paragraph in the discussion about cardiomyopathies mutations:

“The impact of HCM mutations, as well as dilated cardiomyopathy (DCM) mutations can now be evaluated with this accurate high-resolution human β -cardiac IHM structure, including those affecting the MYL2 and MYL3 genes that code for the RLC and ELC proteins, respectively. For example, the restrictive cardiomyopathy (RCM)-E143K ELC mutation⁵¹, the HCM-R58Q⁵² and HCM-D166V RLC mutations^{53,54} correspond respectively to a _{BH}ELC residue and two _{FH}RLC residues of the surface of the IHM that would interact with the core of the thick filament. The IHM structure thus provides the missing puzzle piece to understand the molecular mechanism of inherited cardiomyopathies. A detailed examination of myosin HCM and DCM mutations in relation to our high-resolution structure is beyond the scope of this report and will be the subject of a subsequent publication. The availability of the cardiac IHM structure paves the road towards the rational design of novel therapeutics against distinct families of inherited muscle disease.” (l. 441-451).

The mutations located at the interface are still indicated in Supplementary Table 2.

28. Fig 1f. It might help to have a different color for the NTEs to make them stand out.

This has been corrected. These regions are now represented as “balls”.

29. L518-519. The phosphate is hydrolyzed? The fitting in (c) and (d) is not convincing. It could help to make a figure showing only the map density of the ligands, ions and surrounding/interacting side chains and the corresponding parts of the fitted models and remove the rest. This would make things clearer to the reader. This is done nicely in Heissler Fig 1.

A new figure was added to specifically show the fit of the Mg.ADP.Pi in Supplementary Figure 2a, 2b. There is no ambiguity that the P_i is hydrolyzed.

30. L528. Purple should be magenta.

This has been corrected.

31. L539. The relative positions of FH and BH are not clear from (a). Please change the representation.

Figure 4a has been changed to make the presentation clearer. Note that the comparison with our structure and the different models previously proposed for cardiac myosin are also provided in the same orientation in Supplementary Movie 4 to indicate that previous models could not predict this difference correctly.

32. Some of the domain/loop terminology is not well known (e.g. what is top loop?). It would be very helpful to have a table with such definitions, the amino acids involved, etc. to make navigating the figures easier.

The definition of the top-loop is now added in the legend of Supplementary Figure 4.

“(d) In the FH, the Converter/ELC interface is maintained by “musical chairs”, a set of labile charged residues located on the ELC (deep teal cyan) or in the Converter, more specifically in the Top-loop³ (orange, aa 724-738).” (l. 68-70, Supplementary Material).

33. L549-550. Yang [14] should be cited, as they showed this there.

The reference was added.

34. ED Fig 3. “(e) and (f) compare ... the musical chairs are represented on both panels.” How are they represented? What is the color coding for this figure?

The sentence has been modified as follows:

“(e) and (f) compare the motor/ELC interface and how it is completely altered by the kink present in the BH, the side chains of the musical chairs are represented as sticks on both panels.” (l. 70-72, Supplementary Material).

The color coding was defined in the figure. It is now also defined at the beginning of the legend:

“(a) The blocked head (BH, in green) and the free head (FH, in blue) superimpose on the motor domain with a RMSD of 0.7 Å, the only difference being the kink at the Pliant region in the lever arm, resulting in an angle of 53° between the IQ1 helix of the BH and the FH heads. (b) and (c) show the superimposition of the BH and of the FH respectively with the apo cardiac MD structure in the PPS state (colored in black, MD-PPS-Apo, PDB code 5N6A,²).” (l. 62-66, Supplementary Material).

35. ED Fig 7. What is the color coding?

Now Supplementary Fig.8. The colors are now explicitly stated and they are similar to those defined in Fig. 2a. We are now defining them in legend and in the figure to help the reader. We also add a panel to directly compare the two ELC/RLC interface in this figure, thus making easy to compare them in the same figure. Finally, while the labels in the figure and the legend were explaining the colour codes for panel (c) : (This is specifically true for the region 834-848 of the heavy chain (HC), due to the fact that the coiled-coil triggers a shift in position of the HC residues (arrow between the M849 position). The HC residues (834-848) thus adopt drastically different conformations (yellow for BH, red for FH) that are critical to form the RLC/RLC interface), we have changed those colours to match

the same colours as those used for panel (a) and (b), so that the reader might be less confused.

“Asymmetry in the light chain binding regions. Color coding of the different elements are indicated with labels on the figures (BH: forest green, FH: dark blue, _{FH}ELC: magenta; _{BH}ELC: pink, _{BH}RLC: sand yellow; _{BH}RLC: light grey). (a) Interactions at the ELC/RLC interface in FH (left) differ from the ELC/RLC interface in BH (right). A few side chains involved in polar contacts are shown. (b) Lever arm of the BH and of the FH aligned on IQ2 reveals a 18° difference in the orientation as well as major differences in the ELC/RLC interface. (c) Asymmetry in the conformation and contacts of the two RLCs in the IHM structure. The RLC/IQ2 of the BH and FH heads see a different environment. This is specifically true for the region 834-848 of the heavy chain (HC), due to the fact that the coiled-coil triggers a shift in position of the HC residues (arrow between the M849 position). The HC residues (834-848) thus adopt drastically different conformations (yellow for BH, ruby red for FH) that are critical to form the RLC/RLC interface.” (l. 149-159, Supplementary Material).

36. ED Fig 8. I can see how S2 matches up to the BH mesa. But the labeling on the FH and BH interaction surfaces does not match up. Please fix. Fig 2a footprint does a better job using colors. What is the color coding for the surface charge in ED Fig 8.

Extended Figure 8 has been removed from the new version of the manuscript since it is not necessary anymore.

37. ED Movie 1 and 2 legends. Colors used to define BH and FH (blue and green) are reversed.

We thank Reviewer #3 for pointing this mistake out. This has been corrected.

38. ED Table 1. Map sharpening B-factor (Å²)?

This has been modified.

** See Nature Portfolio's author and referees' website at www.nature.com/authors for information about policies, services and author benefits.

This email has been sent through the Springer Nature Tracking System NY-610A-NPG&MTS

Confidentiality Statement:

This e-mail is confidential and subject to copyright. Any unauthorised use or disclosure of its contents is prohibited. If you have received this email in error please notify our Manuscript Tracking System Helpdesk team at <http://platformsupport.nature.com>.

Details of the confidentiality and pre-publicity policy may be found here

<http://www.nature.com/authors/policies/confidentiality.html>

Privacy Policy | Update Profile

Reviewer #3 (Remarks to the Author)

The revised version of this paper is a big improvement. The authors have responded appropriately to the original critique and the paper now communicates more clearly, focuses on key points that had been treated too briefly before, and removes material (mutations etc.) that were not well justified in the initial submission. I appreciate all the effort that has been put into improving the manuscript and believe it will be a major contribution to the field.

We thank reviewer #3 for the positive appreciation of our new manuscript. A substantial change has been made regarding the final version of the material. The concept of “inter-crown” interface was reincluded in the manuscript. Indeed, regarding removal of materials, we initially agreed to remove since we needed to save space. We do not agree that they were not justified. We have decided to keep the figure related to the inter-crown interface. Even though the docking of our high-resolution structure was done on a low-resolution map of the cardiac myosin filament, it is precisely the high resolution of our structure that allows the arguments that we made regarding the inter-crown interface. A preprint came out recently that describes the same hypothesis (Dutta *et al.*, 2023), where no high-resolution data is presented. To have our paper properly cited as the origin of this concept, our interpretation of the results has been re-included in the manuscript, taking into account the reviewers' technical concerns raised in the first round (radial collapse, Koubassova *et al.*, 2022), which is cited and discussed in the manuscript. Since our hypothesis was formed independent of this preprint, the reference to that work is only in the Peer Review File but not in the paper.

One comment I would make. Rev 2 and Rev 3 both state that the paper at first glance seems to imply that the IHM is not conserved. This goes against the published data and most people's conception of the IHM as a structure that has been around since the earliest animals. What the authors mean is that the (detailed) structure of the IHM is different in different systems. It would be good to emphasize this distinction so that readers don't start to think the overall IHM motif doesn't exist in some systems.

We thank Reviewer #3 for this remark. While this point is now clear in the results and discussions, the end of the introduction could be misleading. The sentence “Our results demonstrate that the IHM structure is not conserved amongst different class 2 myosins. Fundamental differences occur in the lever arm and the head/head interfaces of the cardiac vs. smooth muscle IHM” has now been replaced by “Our study demonstrates how the IHM structure, which is a conserved feature of all class 2 myosins, diverged in different muscle types” (l.89-90).

I have only a few remaining minor comments for the authors to consider for improving clarity and ensuring accuracy of what is written. These are mostly my opinion to help the reader, and the authors should decide whether they want to make changes or not.

L75. “a recent study attempted to use homology modeling based on the SmIHM to predict the effects of HCM mutations on the interfaces stabilizing the off-state of human β -cardiac myosin28.” Please check if this is correct. I don't see it in the Scarff paper myself. The closest I see them come to saying this is: “Given the close agreement between the structure of the IHM in our SmM structure and IHM structures in thick filaments, this structure may also be invaluable as a model for understanding mutations in striated muscle myosins, such as those in β -cardiac myosin that result in hypertrophic cardiomyopathy.”

We changed this part to: “a recent study suggested the use of homology based on the SmIHM to predict the effects of HCM mutations on the interfaces stabilizing the off-state of human β -cardiac myosin”.

LL98-100. Not clear to me why SI Fig2 is shown here. To me it interrupts the flow.

As discussed in the previous rebuttal, we think that this figure is important for the general reader unfamiliar with the Smooth Myo2 IHM.

L106. The only place where the full EM map is shown is in SI Fig 1a. Seems like a wasted opportunity to showcase the beautiful structure.

The full maps (map 1 and 2) are shown in detail in **Supplementary Movie 1 & 2** respectively.

L121-122. Was density for the ELC and RLC N-terminal extensions actually seen in the map?

The N-terminal extensions of the ELC and the RLC are partially seen in the map. This is discussed in the text:

“Finally, two essential regions of regulation, the N-terminal extensions (N-term extensions) of the ELC and RLC, were partially rebuilt in density (Fig. 1f, see Methods).” (l. 120-123)

“Based on our structure, the ELC and the RLC N-terminal extensions range from residues 1-48 and 1-27 respectively, and density allowed us to attribute the conformation of the following residues : BH-ELC : 39-195 ; BH-RLC : 20-163 ; FH-ELC : 39-195 ; FH-RLC : 20-163.” (Methods, l. 496-499).

LL160-163. Just a comment: the superpositions of structures on p. 19 of the rebuttal look nicer than they do in Fig. 2. Worth using the color approach instead of the black/gray used for 5tby in Fig. 2?

We have tried several combinations of color. At the moment the black/white approach is the best in terms of contrast.

L162. “left and center” should be “left and right”?
We thank the Reviewer. This was changed.

L166. It would help to label the Relay.
We thank the Reviewer. The label “Relay” was added in panel Fig. 2g.

L183, 185. I think 5f should be 5g.
We thank the Reviewer. This was fixed.

L236. Would help to label ELC loop 3 on Fig 3.
We thank the Reviewer. The label “Loop-1” was added on Fig. 3.

L249. I think Supplementary Fig. 3a should be Supplementary Fig. 4a.
We thank the Reviewer. This was changed.

LL283-284. In SI Fig 9c,d, I see the closed back door in the BH. But in the FH, E466 does not appear to close the door. This is confusing, when the text states that both FH and BH are closed.

We explain it in the legend of Supplementary 3. The statement is as following:

“The cryo-EM density map is represented as a dark grey mesh (contour 11.95σ). In CarBH, the backdoor is closed without ambiguity, in CarFH, the density of Glu466 is not clear and would be compatible with a closed backdoor (the closed backdoor of 56 CarBH is represented as green residues in (c) to compare the conformation).” (Supplementary information)

We are discussing the Ca position, which is undoubtedly in a closed conformation in both the FH and the BH of CarIHM (**Supplementary Fig. 9c, 9d**). Conversely, in SmIHM , the positions of the Ca of the backdoor are in an open conformation (**Supplementary Fig. 9c, 9d**).

L337. I would say it's the S2/BH interface that differs.

This was changed.

L361-362. “the position of the two phosphorylatable serines, S19 in SmRLC and S15 in CarRLC, is conserved (Fig. 4f).” It would help if S15 were marked in Fig. 4f to make this point clear. It would also be more convincing if the map into which the N-terminal extensions have been fitted were shown.

S15 is represented as a bold “S” colored in red in figure 4f. It is explained in the legend as follows: “The phosphorylation sites (S19 for SmRLC , S15 for CarRLC) are indicated as a red “S”” (l. 766-767).

The map in which the N-terminal extensions have been fitted are seen in **Figure 1** and in **Supplementary Movie 1**. We agree that it is not so easy for the N-terminal extensions which are less defined in density, however, the readers can see the map that will be released to see it in detail.

L399. Should Fig. 5b be Fig. 5a?

We thank the Reviewer. This was fixed.

L409. The significance of Fig. 5c is a little confusing. The IHM looks far above the thick filament backbone (which I take to be the green dashed line). Should be in contact. The actin filament appears to be the same diameter as the thick filament backbone, which is not accurate. What do the black dashed lines signify?

The relative thickness of the actin and thick filament diameter has been corrected. They are introduced mainly to characterize which side of the IHM would be facing the actin filament side or the thick filament side.

The reason to make a distance between the green dash and the IHM is to show more clearly how the IHM surface differ this is why we do not present the green dash line in contact with the IHM. The dark dashed and plain lines are introduced to help readers to compare the two FH in the cardiac IHM and Tarantula IHM.

SI Fig 1. (f) and (g) are reversed in the legend. In (f), what do blue and red colors signify? What are the x and y axes in the 3 squares? What are the 3 views shown?

We thank the Reviewer. This was fixed.

According to the conventions, shown in f are the central slices along the zy plane, xz plane and xy plane in which positive (red) and negative (blue) values correspond to density to be added and subtracted, respectively from the mean density. The values are shown in voxel.

SI L69. Where is cyan in (d)?

The sidechains of the residues located on the ELC are colored in “deep teal cyan” which is the specific name of this color according to pymol.

SI L74. ELC: magenta rather than purple?

The color “magenta” is the name of the color code used by pymol. This is why we labelled it as “magenta”.

SI L151-152. I don't see a left and a right part of the figure in (a).

We thank the Reviewer. This was corrected.

SI L343-344. I could not make any sense of Movie 3 (filename: 382281_1_video_7372180_rqghxx) in relation to SI Fig 8.

Supplementary Movie 3 shows the asymmetry of the RLC/RLC interface. To make it clear, we changed the text in the legend as follow: “See also **Supplementary Movie 3** to visualize the asymmetry of this region in detail”.